# ACCELERATING NON-IID FEDERATED LEARNING VIA HETEROGENEITY-GUIDED CLIENT SAMPLING

## ABSTRACT

Statistical heterogeneity of data present at client devices in a federated learning (FL) system renders the training of a global model in such systems difficult. Particularly challenging are the settings where due to resource constraints only a small fraction of clients can participate in any given round of FL. Recent approaches to training a global model in FL systems with non-IID data have focused on developing client selection methods that aim to sample clients with more informative updates of the model. However, existing client selection techniques either introduce significant computation overhead or perform well only in the scenarios where clients have data with similar heterogeneity profiles. In this paper, we propose HiCS-FL (Federated Learning via Hierarchical Clustered Sampling), a novel client selection method in which the server estimates statistical heterogeneity of a client's data using the client's update of the network's output layer and relies on this information to cluster and sample the clients. We analyze the ability of the proposed techniques to compare heterogeneity of different datasets, and characterize convergence of the training process that deploys the introduced client selection method. Extensive experimental results demonstrate that in non-IID settings HiCS-FL achieves faster convergence and lower training variance than state-of-the-art FL client selection schemes. Notably, HiCS-FL drastically reduces computation cost compared to existing selection schemes and is adaptable to different heterogeneity scenarios.

## 1 INTRODUCTION

The federated learning (FL) framework enables privacy-preserving collaborative training of machine learning (ML) models across a number of devices (clients) by avoiding the need to collect private data stored at those devices. The participating clients typically experience both the system as well as statistical heterogeneity (Li et al., 2020a). The former describes settings where client devices have varying degree of computational resources, communication bandwidth and fault tolerance, while the latter refers to the fact that the data owned by the clients may be drawn from different distributions. In this paper, we focus on FL under statistical heterogeneity and leave studies of system heterogeneity to future work.

An early FL method, FedAvg (McMahan et al., 2017), performs well in the settings where the devices train on independent and identically distributed (IID) data. However, compared to the IID scenario, training on non-IID data is detrimental to the convergence speed, variance and accuracy of the learned model. This has motivated numerous studies aiming to reduce the variance and improve convergence of FL on non-IID data (Karimireddy et al., 2020; Li et al., 2020b). Strategies aiming to tackle statistical heterogeneity in FL can be organized in four categories: (1) adding regularization terms to mitigate objective drift in local training (Chen et al., 2023; Li et al., 2020b); (2) aggregation schemes at the server (Hsu et al., 2019; Wang et al., 2020a); (3) data augmentation by synthesizing artificial data (Hao et al., 2021; Yoon et al., 2021; Chen & Vikalo, 2023); and (4) personalized federated learning (Collins et al., 2021; Fallah et al., 2020; Li et al., 2021b; T Dinh et al., 2020), which allows clients to train customized models rather than a shared global model.

On another note, constraints on communication resources and therefore on the number of clients that may participate in training additionally complicate implementation of FL schemes. It would be particularly unrealistic to require regular contributions to training from all the clients in a large-scale cross-device FL system. Instead, only a fraction of clients participate in any given training round;

unfortunately, this further aggravates detrimental effects of statistical heterogeneity. Selecting informative clients in non-IID FL settings is an open problem that has received considerable attention from the research community (Cho et al., 2020; Fraboni et al., 2021; Goetz et al., 2019). Since privacy concerns typically prohibit clients from sharing their local data label distributions, existing studies focus on estimating informativeness of a client's update by analyzing the update itself. This motivated a family of methods that rely on the norms of local updates to assign probabilities of sampling the clients (Chen et al., 2020; Ribero & Vikalo, 2020). Aiming to enable efficient use of the available communication and computation resources, another set of methods groups clients with similar data distributions into clusters based on the similarity between clients' model updates (Balakrishnan et al., 2022; Fraboni et al., 2021). In addition to model updates, local loss (Cho et al., 2020; Lai et al., 2021; Tang et al., 2022) and test accuracy on public data (Kim et al., 2020; Mohammed et al., 2020) have been used to gauge informativeness of the clients' updates. Across the board, the existing methods still struggle to deliver desired performance in an efficient manner: the loss-based and clustering-based techniques cannot distinguish clients with balanced data from the clients with imbalanced data, while accuracy-based methods rely on a validation dataset at the server and need to train a sampling agent using multi-armed bandit (MAB) algorithms.

In this paper, we consider training a neural network model for classification tasks via federated learning and propose a novel adaptive clustering-based sampling method for identifying and selecting informative clients. The method, referred to as Federated Learning via Hierarchical Clustered Sampling (HiCS-FL), relies on the updates of the (fully connected) output layer in the network to determine how diverse is the clients' data and, based on that, decide which clients to sample. In particular, HiCS-FL enables heterogeneity-guided client selection by utilizing general properties of the gradients of the output layer to distinguish between clients with balanced from those with imbalanced data. Unlike the Clustered Sampling strategies (Fraboni et al., 2021) where the clusters of clients are sampled uniformly, HiCS-FL allocates different probabilities (importance) to the clusters according to their average estimated data heterogeneity. Numerous experiments conducted on FMNIST (Xiao et al., 2017), CIFAR10 and CIFAR100 demonstrate that HiCS-FL achieves significantly faster training convergence and lower variance than the competing methods. Finally, we conduct convergence analysis of HiCS-FL and discuss implications of the results.

In summary, the contributions of the paper include: (1) Analytical characterization of the correlation between local updates of the output layer and the FL clients' data label distribution, along with an efficient method for estimating data heterogeneity; (2) a novel clustering-based algorithm for heterogeneity-guided client selection; (3) extensive simulation results demonstrating HiCS-FL provides significant improvement in terms of convergence speed and variance over competing approaches; and (4) theoretical analysis of the proposed schemes.

## 2 BACKGROUND AND RELATED WORK

Assume the cross-device federated learning setting with $N$ clients, where client $k$ owns private local dataset $\mathcal{B}_k$ with $|\mathcal{B}_k|$ samples. The plain vanilla FL considers the objective

$$\min_{\theta} F(\theta) \triangleq \sum_{k=1}^{N} p_k F_k(\theta), \tag{1}$$

where $\theta$ denotes parameters of the global model, $F_k(\theta)$ is the loss (empirical risk) of model $\theta$ on $\mathcal{B}_k$, and $p_k$ denotes the weight assigned to client $k$, $\sum_{k=1}^{N} p_k = 1$. In FedAvg, the weights are set to $p_k = |\mathcal{B}_k| / \sum_{i=1}^{N} |\mathcal{B}_i|$. In training round $t$, the server collects clients' model updates $\theta_k^t$ formed by training on local data and aggregates them to update global model as $\theta^{t+1} = \sum_{k=1}^{N} p_k \theta_k^t$.

When an FL system operates under resource constraints, typically only $K \ll N$ clients are selected to participate in any given round of training; denote the set of clients selected in round $t$ by $\mathcal{S}^t$. In departure from FedAvg, FedProx (Li et al., 2020b) proposes an alternative strategy for sampling clients based on a multinomial distribution where the probability of selecting a client is proportional to the size of its local dataset; the global model is then formed as the average of the collected local models $\theta^{t+1} = \frac{1}{K} \sum_{k \in \mathcal{S}^t} \theta_k^t$. This sampling strategy is *unbiased* since the the updated global model is on expectation equal to the one obtained by the framework with full client participation 1.

AFL (Goetz et al., 2019) is the first study to utilize local validation loss as a *value* function for computing client sampling probabilities; Power-of-Choice (Cho et al., 2020) takes a step further to

propose a greedy approach to sampling clients with the largest local loss. Both of these methods require all clients to compute the local validation loss, which is often unrealistic. To address this problem, FedCor (Tang et al., 2022) models the local loss by a Gaussian Process (GP), estimates the GP parameters from experiments, and uses the GP model to predict clients' local losses without requiring them to perform validation; unfortunately, FedCor is an empirical scheme that does not come with theoretical convergence analysis/guarantees. In (Chen et al., 2020), Optimal Client Sampling scheme aiming to minimize the variance of local updates by assigning sampling probabilities proportional to the Euclidean norm of the updates is proposed. The study in (Ribero & Vikalo, 2020) models the progression of model's weights by an Ornstein-Uhlenbeck process and proposes a strategy, optimal under that assumption, for selecting clients with significant weight updates.

The clustering-based sampling method proposed in (Fraboni et al., 2021) uses cosine similarity (Sattler et al., 2020) to group together clients with similar local updates, and proceeds to sample one client per cluster in attempt to avoid redundant gradient information. DivFL (Balakrishnan et al., 2022) follows the same principle of identifying representative clients but does so by constructing a submodular set and greedily selecting diverse clients. Both of these techniques are computationally expensive due to the high dimension of the gradients that they need to process. Another line of work (Shi et al., 2023; Yang et al., 2021b; Zhang et al., 2022) relies on public data at the server or pairwise inner products of clients' data label distributions to design MAB algorithms that assign sampling probabilities leading to subsets of clients with low class-imbalance degree.

In general, the overviewed methods either: (1) select diverse clients to reduce redundant information; or (2) select clients with a perceived significant contributions to the global model (high loss, large update or low class-imbalance). Efficient and effective client selection in FL remains an open challenge, motivating the heterogeneity-guided adaptive client selection method presented next.

## 3  HiCS-FL: FL via Hierarchical Clustered Sampling

Existing client sampling methods including Clustered Sampling (Fraboni et al., 2021) and DivFL (Balakrishnan et al., 2022) aim to select clients such that the resulting model update is an unbiased estimate of the true update (i.e., the update in the case of full client participation) while minimizing the variance

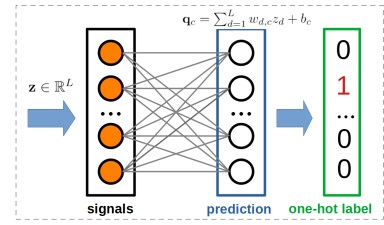

$$\left\| \frac{1}{N} \sum_{k=1}^{N} \nabla F_k(\theta^t) - \frac{1}{K} \sum_{k \in \mathcal{S}^t} \nabla F_k(\theta^t) \right\|_2^2. \quad (2)$$

Figure 1: The last two network layers.

Clustered Sampling, for instance, groups $N$ clients into $K$ clusters based on *representative gradients* (Sattler et al., 2020), and randomly selects one client from each cluster to contribute to the global model update. Such an approach unfortunately fails to differentiate between model updates formed on data with balanced and those formed on data with imbalanced label distributions – indeed, in either case the updates are treated as being equally important. However, a number of studies in centralized learning has shown that class-imbalanced datasets have significant detrimental effect on the performance of learning classification tasks (Buda et al., 2018; Chawla et al., 2002; Shen et al., 2016). This intuition carries over to the FL settings where one expects the updates from clients training on relatively more balanced local data to have a more beneficial impact on the performance of the system. The Federated Learning via Hierarchical Clustered Sampling (HiCS-FL) framework described in this section adapts to the clients' data heterogeneity in the following way: if the levels of heterogeneity (as quantified by the entropy of data label distribution) vary from one cluster to another, HiCS-FL is more likely to sample clusters containing clients with more balanced data; if the clients grouped in different clusters have similar heterogeneity levels, HiCS-FL is more likely to select diverse clients (i.e., sample uniformly across clusters, thus reducing to the conventional clustered sampling strategy).

### 3.1  Class-imbalance causes objective drift

A number of studies explored detrimental effects of non-IID training data on the performance of a global model learned via FedAvg. An example is SCAFFOLD (Karimireddy et al., 2020) which demonstrates *objective drift* in non-IID FL manifested through large differences between local models $\theta_k^*$ trained on substantially different data distributions. The drift is due to FedAvg updating the

global model in the direction of the weighted average of local optimal models, which is not necessarily leading towards the optimal global model $\theta^*$. The optimal model $\theta^*$, in principle obtained by solving optimization in Eq. 1, achieves minimal empirical error on the data with uniform label distribution and is intuitively closer to the local optimal models trained on balanced data. Recent work (Zhang et al., 2022) empirically verified this conjecture through extensive experiments. Let $\nabla F(\theta^t)$ denote the gradient of $F(\theta^t)$ given the global model $\theta^t$ at round $t$; the difference between $\nabla F(\theta^t)$ and the local gradient $\nabla F_k(\theta^t)$ computed on client $k$'s data is typically assumed to be bounded (Chen et al., 2020; Fraboni et al., 2021; Wang et al., 2020b). To proceed, we formalize the assumption about the relationship between gradients and data label distributions.

**Assumption 1 (Bounded Dissimilarity.)** *Gradient $\nabla F_k(\theta^t)$ of the k-th local model at global round $t$ is such that*

$$\left\| \nabla F_k(\theta^t) - \nabla F(\theta^t) \right\|^2 \leq - \exp \left( \beta \left[ H(\mathcal{D}^{(k)}) - H(\mathcal{D}_0) \right] \right) \rho + \kappa = \sigma_k^2, \tag{3}$$

*where $\mathcal{D}^{(k)}$ is the data label distribution of client $k$, $\mathcal{D}_0$ denotes uniform distribution, $H(\cdot)$ is Shannon's entropy of a stochastic vector, and $\beta > 0, \kappa > \rho > 0$.*

The assumption commonly encountered in literature (Chen et al., 2020; Fraboni et al., 2021; Wang et al., 2020b) is recovered by setting the right-hand side of (3) to $\sigma_m^2 = \max_k \sigma_k^2$. Intuitively, if the data label distribution of client $k$ is highly imbalanced (i.e., $H(\mathcal{D}^{(k)})$ is small), the local gradient $\nabla F_k(\theta^t)$ may significantly differ from the global gradient $\nabla F(\theta^t)$ (as reflected by the bound above). Analytically, connecting the gradients to the local data label distributions allows one to characterize the effects of client selection on the variance and the rate of convergence. The results of extensive experiments that empirically verify the above assumption are reported in Appendix A.2.

---

**Algorithm 1** HiCS-FL

**Input:**
    Datasets distributed across $N$ clients, the number of clients to sample $K$, total global rounds $\mathcal{T}$.
1: Initialize updates of bias $\Delta \mathbf{b}^{(k)} \leftarrow \mathbf{0} \ \forall k \in [N]$, global model $\theta^t \leftarrow \theta^1$, $S_0 = [N]$.
2: **for** $t = 1, \ldots, \mathcal{T}$ **do**
3:   **if** $t \leq \lceil N/K \rceil$ **then**
4:     $\mathcal{S}^t \leftarrow$ randomly sample $\min(K, |S_0|)$ clients from $S_0$, update $S_0 \leftarrow S_0 - \mathcal{S}^t$;
5:   **else**
6:     estimate $\hat{H}^t(\mathcal{D}^{(k)}), \forall k \in [N]$ and cluster $N$ clients into $M$ groups based on Eq. 9;
7:     $\mathcal{S}^t \leftarrow \emptyset$;
8:     **while** $|\mathcal{S}^t| < K$ **do**
9:       sample group $G_m^t$ according to $\pi^t$;
10:       sample client $k$ in $G_m^t$ according to $\tilde{\mathbf{p}}_m$;
11:       $\mathcal{S}^t \leftarrow \mathcal{S}^t \cup k$;
12:     **end while**
13:   **end if**
14:   **for** $k \in \mathcal{S}^t$ **do**
15:     $\theta_k^t \leftarrow$ **LocalUpdate**$(\theta^t), \Delta \mathbf{b}^{(k)} \in \theta_k^t - \theta^t$
16:   **end for**
17:   $\theta^{t+1} \leftarrow \frac{1}{K} \sum_{k \in \mathbf{S}^t} \theta_k^t$;
18:   $\Delta \mathbf{b}^{(k)} \leftarrow \Delta \mathbf{b}^{(k)}, \forall k \in \mathcal{S}^t$;
19: **end for**
**Output:**
    The global model $\theta^{\mathcal{T}+1}$

---

## 3.2 ESTIMATING CLIENT'S DATA HETEROGENEITY

If the server were given access to clients' data label distributions, selecting clients would be relatively straightforward (Wolfrath et al., 2022). However, privacy concerns typically discourage clients from sharing such information. Previous studies have explored the use of multi-arm bandits for inferring clients' data heterogeneity from local model parameters, or have utilized a validation dataset at the server to accomplish the same (Shi et al., 2023; Yang et al., 2021b; Zhang et al., 2022). In this section, we demonstrate how to efficiently and accurately estimate data heterogeneity using local updates of the output layer of a neural network in a classification task. Figure 1 illustrates the last two layers in a typical neural network. The prediction $\mathbf{q} \in \mathbb{R}^C$ is computed by forming a weighted average of signals $\mathbf{z} \in \mathbb{R}^L$ utilizing the weight matrix $\mathbf{W} \in \mathbb{R}^{C \times L}$ and bias $\mathbf{b} \in \mathbb{R}^C$.

### 3.2.1 LOCAL UPDATES OF THE OUTPUT LAYER

An empirical investigation of the gradients of the output layer's weights while training with FedAvg using mini-batch stochastic gradient descent (SGD) as an optimizer is reported in (Wainakh et al., 2021). There, the focus is on detecting the presence of specific labels in a batch rather than on exploring the effects of class imbalance on the local update. To pursue the latter, we focus on the

correlation between local updates of the output layer's bias and the client's data label distribution; we start by analyzing the training via FedAvg that employs SGD and then extend the results to other FL algorithms that utilize optimizers beyond SGD. We assume that the model is trained by minimizing the cross-entropy (CE) loss over one-hot labels – a widely used multi-class classification framework. The gradient is computed by averaging contributions of the samples in mini-batches, i.e., $\nabla_{\mathbf{b}}\mathcal{L}_{\mathbf{CE}} = \frac{1}{Bl}\sum_{j=1}^{l}\sum_{n=1}^{B}\nabla_{\mathbf{b}}\mathcal{L}_{\mathbf{CE}}^{(j,n)}(\mathbf{x}^{(j,n)}, y^{(j,n)})$, where $B$ denotes the batch size, $l$ is the number of mini-batches, $\mathbf{x}^{(j,n)}$ is the $n$-th point in the $j$-th mini-batch and $y^{(j,n)} \in [C]$ is its label. The contribution of $\mathbf{x}^{(j,n)}$ to the $i$-th component of the gradient of the output layer's bias $\mathbf{b}$ can be found as (details provided in Appendix A.3)

$$\nabla_{b_i}\mathcal{L}_{\mathbf{CE}}^{(j,n)} = \mathbb{I}\{i = y^{(j,n)}\}\frac{-\sum_{c\neq i}\exp(q_c^{(j,n)})}{\sum_{c=1}^{C}\exp(q_c^{(j,n)})} + \mathbb{I}\{i \neq y^{(j,n)}\}\frac{\exp(q_i^{(j,n)})}{\sum_{c=1}^{C}\exp(q_c^{(j,n)})}, \tag{4}$$

where $\mathbb{I}\{\cdot\}$ is an indicator, $\mathbf{q}^{(j,n)} = [q_1^{(j,n)}, \ldots, q_C^{(j,n)}]^T = \mathbf{W}\cdot\mathbf{z}^{(j,n)} + \mathbf{b}$ is the output logit for signals $\mathbf{z}^{(j,n)} \in \mathbb{R}^L$ corresponding to training point $(\mathbf{x}^{(j,n)}, y^{(j,n)})$ (see Fig. 1), and where $C$ denotes the number of classes. We make the following observations: (1) the sign of $y^{(j,n)}$-th component of $\nabla_{\mathbf{b}}\mathcal{L}_{\mathbf{CE}}^{(j,n)}$ is opposite of the sign of other components; and (2) the $y^{(j,n)}$-th component of $\nabla_{\mathbf{b}}\mathcal{L}_{\mathbf{CE}}^{(j,n)}$ is equal in magnitude to all the other components combined. Note that the above two observations are standard for neural networks using CE loss for supervised multi-class classification tasks.

In each global round $t$ of FedAvg, the selected client $k$ starts from the global model $\theta^t$ and proceeds to compute local update in $R$ local epochs employing an SGD optimizer with learning rate $\eta$. According to Eq. 4, the $i$-th component of local update $\Delta\mathbf{b}^{(k)}$ is computed as

$$\Delta b_i^{(k)} = -\frac{\eta}{Bl}\sum_{j=1}^{l}\sum_{n=1}^{B}\sum_{r=1}^{R}\nabla_{b_i}\mathcal{L}_{\mathbf{CE}}^{(j,n,r)}, \tag{5}$$

where $\nabla_{b_i}\mathcal{L}_{\mathbf{CE}}^{(j,n,r)}$ denotes the gradient of bias at local epoch $r$. Note that the local update of client $k$, $\Delta\mathbf{b}^{(k)}$, is dependent on the label distribution of client $k$'s data, $\mathcal{D}^{(k)} = [D_1^{(k)}, \ldots, D_C^{(k)}]^T$ and the label-specific components of $\mathbf{q}^{(j,n)}$ which change during training. We proceed by relating expected local updates to the label distributions; for convenience, we first introduce the following definition.

**Definition 1** *Let $\mathcal{B}^{-i}$ be the subset of local data $\mathcal{B}$ that excludes points with label $i$. Let $\mathbf{s}^{-i}(\mathbf{x}) \in [0,1]^C$ be the softmax output of a trained neural network for a training point $(\mathbf{x}, y) \in \mathcal{B}^{-i}$. The $i$-th component of $\mathbf{s}^{-i}(\mathbf{x})$, $s_i^{-i}(\mathbf{x})$, indicates the level of confidence in (erroneously) classifying $\mathbf{x}$ as having label $i$. For convenience, we define $\mathcal{E}_i = \mathbb{E}_{(\mathbf{x},y)\sim\mathcal{B}^{-i}}\left[\mathbf{s}_i^{-i}(\mathbf{x})\right], \forall i \in [C]$.*

In an untrained/initialized neural network where classifier makes random predictions, $\mathcal{E}_i = 1/C$; as training proceeds, $\mathcal{E}_i$ decreases. By taking expectation and simplifying, we obtain (details provided in Appendix A.4)

$$\mathbb{E}\left[\Delta b_i^{(k)}\right] = \eta R\left(D_i^{(k)}\sum_{c=1}^{C}\mathcal{E}_c - \mathcal{E}_i\right), \tag{6}$$

where $D_i^{(k)}$ denotes the true fraction of samples with label $i$ in client $k$'s data, $\sum_{i=1}^{C}D_i^{(k)} = 1$.

### 3.2.2 ESTIMATING LOCAL DATA HETEROGENEITY

We quantify the heterogeneity of clients' data by an entropy-like measure defined below. Let $\mathcal{D}^{(k)}$ denote the label distribution of client $k$'s data; its entropy is defined as $H(\mathcal{D}^{(k)}) \triangleq -\sum_{i=1}^{C}D_i^{(k)}\ln D_i^{(k)} \leq \ln C$. Recall that more balanced data results in higher entropy, and that $H(\mathcal{D}^{(k)})$ takes the maximal value when $\mathcal{D}^{(k)}$ is uniform. The server does not know $\mathcal{D}^{(k)}$ and therefore cannot compute $H(\mathcal{D}^{(k)})$ directly. We define

$$\hat{H}(\mathcal{D}^{(k)}) \triangleq H(\text{softmax}(\Delta\mathbf{b}^{(k)}, T)), \tag{7}$$

where $\text{softmax}(\Delta\mathbf{b}^{(k)}, T)_i = \exp(\Delta b_i^{(k)}/T)/\sum_{c=1}^{C}\exp(\Delta b_c^{(k)}/T)$, $1 \leq i \leq C$; here $T$ is a scaling hyper-parameter (so-called *temperature*). Note that even though we can compute $\hat{H}(\mathcal{D}^{(k)})$ to characterize heterogeneity, $D_i^{(k)}$ and $\mathcal{E}_i$ remain unknown to the server (details in Appendix A.5).

**Theorem 1** *Consider an FL system in which clients collaboratively train a model for a classification task over $C$ classes. Let $\mathcal{D}^{(u)}$ and $\mathcal{D}^{(k)}$ denote data label distributions of an arbitrary pair of clients $u$ and $k$, respectively. Moreover, let $\mathbf{U}$ denote the uniform distribution, and let $\eta$ and $R$ be the learning rate and the number of local epochs, respectively. Then*

$$\mathbb{E}\left[\hat{H}(\mathcal{D}^{(u)}) - \hat{H}(\mathcal{D}^{(k)})\right] \geq \frac{1}{2}\left(\frac{\eta R}{CT}\sum_{c=1}^{C}\mathcal{E}_c\right)^2 \left\|\mathcal{D}^{(k)} - \mathbf{U}\right\|_2^2 - \frac{\eta R}{T}\left\|\mathcal{D}^{(u)} - \mathbf{U}\right\|_\infty - \mathcal{C}\delta, \quad (8)$$

*where $\mathcal{C} = \frac{\eta R(\eta R + C^2 T \ln C)}{C^2 T^2}$ and $\delta = \max_i \left|\frac{\sum_{c=1}^{C}\mathcal{E}_c}{C} - \mathcal{E}_i\right|$.*

The proof is provided in Appendix A.6. As an illustration, consider the scenario where client $u$ has a balanced dataset while the dataset of client $k$ is imbalanced; then $\|\mathcal{D}^{(k)} - \mathbf{U}\|_2^2$ is relatively large compared to $\|\mathcal{D}^{(u)} - \mathbf{U}\|_\infty$. The bound in (8) also depends on $\delta$, which is reflective of how misleading on average can a class be; small $\delta$ suggests that no class is universally misleading. As shown in Appendix A.4, during training $\delta$ gradually decreases to 0 as $\sum_{i=1}^{C}\mathcal{E}_i$ decreases to 0.

### 3.2.3 GENERALIZING BEYOND FEDAVG AND SGD

The proposed method for estimating clients' data heterogeneity relies on the properties of the gradient for the cross-entropy loss objective discussed in Section 3.2.1. However, for FL algorithms other than FedAvg, such as FedProx (Li et al., 2020b), FedDyn (Acar et al., 2021) and Moon (Li et al., 2021a), which add regularization to combat overfitting, the aforementioned properties may not hold. Moreover, optimization algorithms using second-order momentum such as Adam (Kingma & Ba, 2014) deploy update rules different from SGD, making the local updates no longer proportional to the gradients. Nevertheless, HiCS-FL remains capable of distinguishing between clients with imbalanced and balanced data. Further discussion of various FL algorithms with optimizers beyond SGD are in appendix A.8 and A.9.

### 3.3 HETEROGENEITY-GUIDED CLUSTERING

Clustered Sampling (CS) (Fraboni et al., 2021) uses cosine similarity (Sattler et al., 2020) between gradients to quantify proximity between clients' data distributions and subsequently group them into clusters. However, cosine similarity cannot help distinguish between clients with balanced and those with imbalanced datasets. Motivated by this observation, we introduce a new distance measure that incorporates estimates of data heterogeneity $\hat{H}(\mathcal{D}^{(k)})$. In particular, the proposed measure of distance between clients $u$ and $k$ that we use to form clusters is defined as

$$\textbf{Distance}(u, k) = \lambda \arccos\left(\frac{\Delta \mathbf{b}^{(u)} \cdot \Delta \mathbf{b}^{(k)}}{|\Delta \mathbf{b}^{(u)}| \cdot |\Delta \mathbf{b}^{(k)}|}\right) + (1 - \lambda)\left|\hat{H}(\mathcal{D}^{(u)}) - \hat{H}(\mathcal{D}^{(k)})\right|, \quad (9)$$

where the first term is akin to the cosine similarity used by CS with the major difference that we compute it using only the updates of the bias in the output layer, which is much more efficient than using the weights of the entire network; $\lambda$ is a pre-defined hyper-parameter (set to $0.1$ in all our experiments). For small $\lambda$, the second term dominates when there are clients with different levels of statistical heterogeneity; this allows emergence of clusters that group together clients with balanced datasets. The second term is small when clients have data with similar levels of statistical heterogeneity; in that case, the distance measure reduces to the conventional cosine similarity.

### 3.4 HIERARCHICAL CLUSTERED SAMPLING

To select $K$ out of $N$ clients in an FL system, we first organize the clients into $M \geq K$ groups via the proposed Hierarchical Clustered Sampling (HiCS) technique. In particular, during the first $\lceil N/K \rceil$ training rounds the server randomly (without replacement) selects clients and collects from them local updates of $\Delta \mathbf{b}^{(k)}$; the server then estimates $\hat{H}^t(\mathcal{D}^{(k)})$ for each selected client $k$ and clusters the clients using the distance measure defined in Eq. 9. Let $G_1^t, \ldots, G_M^t$ denote the resulting $M$ clusters at global round $t$, and let $\bar{H}_m^t = \frac{1}{|G_m|}\sum_{k \in G_m}\hat{H}^t(\mathcal{D}^{(k)})$ characterize the average heterogeneity of clients in cluster $m$, $m \in [M]$. Having computed $\bar{H}_m^t$, HiCS selects a cluster according to the probability vector $\pi^t$, and then from the selected cluster selects a client according to the probability

vector $\tilde{\mathbf{p}}_m^t$. The two probability vectors $\pi^t$ and $\tilde{\mathbf{p}}_m^t$ are defined as

$$\pi^t = \left[ \frac{\exp(\gamma^t \bar{H}_1^t)}{\sum_{m=1}^M \exp(\gamma^t \bar{H}_m^t)}, \ldots, \frac{\exp(\gamma^t \bar{H}_M^t)}{\sum_{m=1}^M \exp(\gamma^t \bar{H}_m^t)} \right], \quad \tilde{\mathbf{p}}_m^t = \left[ \frac{p_{k_1}}{\sum_{k \in G_m} p_k}, \ldots, \frac{p_{k_{|G_m|}}}{\sum_{k \in G_m} p_k} \right], \tag{10}$$

where $k_1, \ldots, k_{|G_m|}$ are the indices of clients in cluster $G_m$, $\gamma^t = \gamma^0(1 - \frac{t}{\mathcal{T}})$ denotes an annealing hyper-parameter, and $\mathcal{T}$ is the number of global rounds. The annealing parameter is scheduled such that at first it promotes sampling clients with balanced data, thus accelerating and stabilizing the convergence of the global model. To avoid overfitting potentially caused by repeatedly selecting a small subset of clients, the annealing parameter is gradually reduced to $\gamma^t \approx 0$, when the server samples the clusters uniformly. The described procedure is formalized as Algorithm 1.

## 3.5 Convergence analysis

Adopting the standard assumptions of smoothness, unbiased gradients and bounded variance (Chen et al., 2020), the following theorem holds for FedAvg with SGD optimizer.

**Theorem 2** *Assume $F_k(\cdot)$ is L-smooth for all $k \in [N]$. Let $\theta^t$ denote parameters of the global model and let $F(\cdot)$ be defined as in Eq. 1. Furthermore, assume the stochastic gradient estimator $g_k(\theta^t)$ is unbiased and the variance is bounded such that $\mathbb{E} \left\| g_k(\theta^t) - \nabla F_k(\theta^t) \right\|^2 \leq \sigma^2$. Let $\eta$ and $R$ be the learning rate and the number of local epochs, respectively. If the learning rate is such that $\eta \leq \frac{1}{8LR}, R \geq 2$, then*

$$\min_{t \in [\mathcal{T}]} \left\| \nabla F(\theta^t) \right\|^2 \leq \frac{1}{\mathcal{T}} \left( \frac{F(\theta^0) - F(\theta^*)}{\mathcal{A}_1} + \mathcal{A}_2 \sum_{t=0}^{\mathcal{T}-1} \sum_{k=1}^N \omega_k^t \sigma_k^2 \right) + \mathbf{\Phi}, \tag{11}$$

*where $\mathcal{A}_1$, $\mathcal{A}_2$, $\mathbf{\Phi}$ are positive constants, and $\omega_k^t$ is the probability of sampling client $k$ at round $t$.*

Note that only the second term in the parenthesis on the right-hand side of the bound in Theorem 2 is related to the sampling method $\Pi$. Under Assumption 1,

$$\sum_{k=1}^N \omega_k^t \sigma_k^2 \leq -\sum_{k=1}^N \omega_k^t \frac{\exp\left(\beta H(\mathcal{D}^{(k)})\right)}{\exp\left(\beta H(\mathcal{D}_0)\right)} \rho + \kappa = -\mathcal{H}_\Pi + \kappa. \tag{12}$$

If the server samples clients with weights proportional to $p_k$, the statistical heterogeneity of the entire FL system may be characterized by $\mathcal{H}_M = \sum_{k=1}^N p_k \frac{\exp(\beta(H(\mathcal{D}^{(k)})))}{\exp(\beta(H(\mathcal{D}_0)))} \rho$. If all clients have class-imbalanced data, $\mathcal{H}_M$ is small and thus random sampling leads to unsatisfactory convergence rate (as indicated by Theorem 2). On the other hand, since the clients sharing a cluster have similar data entropy, the proposed HiCS-FL leads to $\omega_k^t = \frac{p_k \exp(\gamma^t \hat{H}^t(\mathcal{D}^{(k)}))}{\sum_{j=1}^N p_j \exp(\gamma^t \hat{H}^t(\mathcal{D}^{(j)}))}$. When training starts, $\mathcal{H}_\Pi$ is large because the server tends to sample clients with higher $p_k \exp(\gamma^t H(\mathcal{D}^{(k)}))$; as $\gamma^t$ decreases, $\mathcal{H}_\Pi$ eventually approaches $\mathcal{H}_M$. Further details and the proof of the theorem are in Appendix A.7.

## 4 Experiments

**Setup.** We evaluate the proposed HiCS-FL algorithm on three benchmark datasets (FMNIST, CIFAR10 and CIFAR100) using different CNN architectures. We use four baselines: random sampling, pow-d (Cho et al., 2020), clustered sampling (CS) (Fraboni et al., 2021) and DivFL (Balakrishnan et al., 2022). To generate non-IID data partitions, we follow the strategy in (Yurochkin et al., 2019), utilizing Dirichlet distribution with different concentration parameters $\alpha$ which controls the level of heterogeneity (smaller $\alpha$ leads to generating less balanced data). In a departure from previous works we utilize several different $\alpha$ to generate data partitions for a single experiment, leading to a realistic scenario of varied data heterogeneity across different clients. To quantify the performance of the tested methods, we use two metrics: (1) average training loss and variance, and (2) test accuracy of the learned global model. For better visualization, data points in the results are smoothened by a Savitzky–Golay filter with window length 13 and the polynomial order set to 3. Further details of the experimental setting and a visualization of data partitions are in Appendix A.1 and A.10.

### 4.1 AVERAGE TRAINING LOSS AND VARIANCE

**FMNIST.** We run FedAvg with SGD to train a global model in an FL system with 50 clients, where 10% of clients are selected to participate in each round of training. The data partitions are generated using one of 3 sets of the concentration parameter $\alpha$ values: (1) $\{0.001, 0.002, 0.005, 0.01, 0.5\}$; (2) $\{0.001, 0.002, 0.005, 0.01, 0.2\}$; (3) $\{0.001\}$. These are used to generate clients' data so as to emulate the following scenarios: (1) 80% of clients have severely imbalanced data while the remaining 20% have balanced data; (2) 80% clients have severely imbalanced data while the remaining 20% have mildly imbalanced data; (3) all clients have severely imbalanced data. Note that $\mathcal{H}_M$ monotonically decreases as we go through settings (1) to (3). For a fair comparison, pow-d and DivFL are deployed with their ideal settings where the server requires all clients to precompute in each round a metric that is then used for client selection. Figure 2 shows that HiCS-FL outperforms other methods across different settings, exhibiting the fastest convergence rates and the least amount of variance. Particularly significant is the acceleration of convergence in setting (1) where 20% of the participating clients have balanced data. Figure 4(a) shows that when all clients have severely imbalanced data, HiCS-FL performs similarly to (Fraboni et al., 2021) (as expected, see Section 3.3), helping achieve significant reduction of training variations as evident by a smooth loss trajectory.

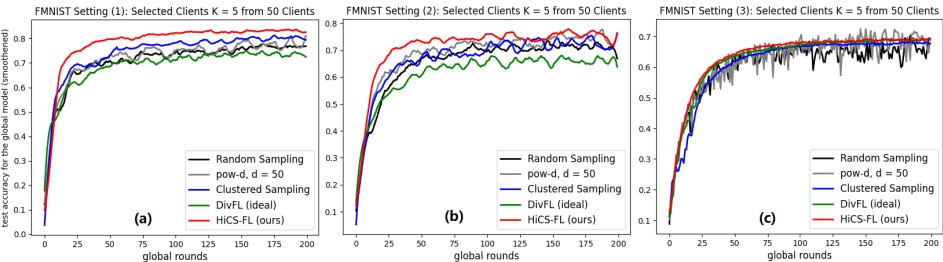

Figure 2: Test accuracy for the global model on 3 groups of data partitions of FMNIST dataset.

**CIFAR10.** Here we compare the performance of HiCS-FL to FedProx (Li et al., 2020b) running Adam optimizer (Kingma & Ba, 2014) on the task of training an FL system with 50 clients, where 20% of clients are selected to participate in each training round. Similar to the experiments on FMNIST, 3 sets of the concentration parameter $\alpha$ are considered: (1) $\{0.001, 0.01, 0.1, 0.5, 1\}$; (2) $\{0.001, 0.002, 0.005, 0.01, 0.5\}$; (3) $\{0.001, 0.002, 0.005, 0.01, 0.1\}$. The interpretation of the scenarios emulated by these setting is as same as in the FMNIST experiments. Figure 3 demonstrates improvement of HiCS-FL over all the other methods. HiCS-FL exhibits particularly significant improvements in settings (2) and (3), where 80% of the clients with extremely imbalanced data benefit from 20% of the clients with either balanced or mildly imbalanced data. The advantage of HiCS-FL in setting (1) where all clients have relatively high data heterogeneity is relatively modest (see Fig. 3) because the system's $\mathcal{H}_M$ is relatively large (see discussion in Section 3.5). It is worth pointing out that in Figure 4(b) the variance of HiCS-FL increases because HiCS-FL gradually reduces $\gamma^t$, eventually becoming plain vanilla random sampling.

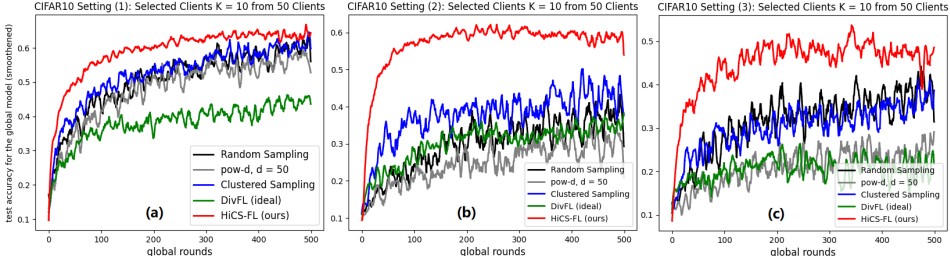

Figure 3: Test accuracy of the global model on three data partitions of CIFAR10 dataset.

**CIFAR100.** As in the CIFAR10 experiments, we compare HiCS-FL to FedProx running Adam optimizer but now consider training of an FL system with 100 clients, where 20% of the clients are selected to participate in each round of training. We consider two settings of the concentration parameter $\alpha$: (1) $\{0.001, 0.01, 0.1, 0.5, 1\}$ and (2) $\{0.001, 0.005, 0.01, 0.1, 1\}$. Setting (1) emulates the scenario where clients have a range of heterogeneity profiles, from extremely imbalanced, through mildly imbalanced, to balanced, while setting (2) corresponds to the scenario where 80% of

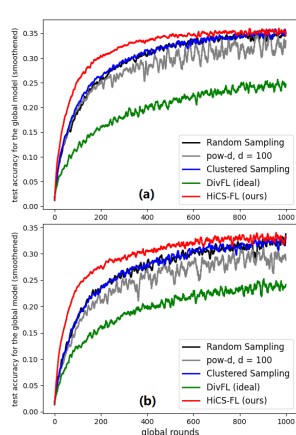

Figure 4: Average training loss and standard deviation of HiCS-FL compared to four baselines for setting (3) on FMNIST and CIFAR10 datasets. More results provided in Appendix A.12.

the clients have extremely imbalanced data while the remaining $20\%$ have balanced data. The system's $\mathcal{H}_{\mathbf{M}}^{(1)}$ for setting (1) is larger than $\mathcal{H}_{\mathbf{M}}^{(2)}$ for setting (2), which is reflected in a more significant improvements achieved by HiCS-FL in the latter setting, as shown in Figure 5.

## 4.2 ACCELERATING THE TRAINING CONVERGENCE

In this section we report the communication costs required to achieve convergence when using HiCS-FL, and compare those results with the competing schemes. For brevity, we select one result from each experiment conducted on the considered three datasets, and display them in Table 1. As can be seen from the table, HiCS-FL significantly reduces the number of communication rounds needed to reach target test accuracy. On FMNIST, HiCS-FL needs $60$ rounds to reach test accuracy $0.75$, achieving it $2.5$ times faster than the random sampling scheme. On CIFAR10, HiCS-FL requires only $123$ rounds to reach $0.6$ test accuracy, which is $7.3$ times faster than random sampling. Acceleration on CIFAR100 is relatively modest but HiCS-FL still outperforms other methods, and does so up to $1.63$ times faster than random sampling.

Figure 5: CIFAR100 acc.

Table 1 also shows that HiCS-FL provides the reported improvements without introducing major computational and communication overhead. The only additional computation is due to estimating data heterogeneity and performing clustering utilizing bias updates, which scales with the total number of classes but does not increase with the size of the neural network model $|\theta^t|$. Remarkably, HiCS-FL outperforms pow-d, Clustered Sampling and DivFL in terms of convergence speed, variance and test accuracy while requiring significantly less computations. More details are provided in Appendix A.13.

Table 1: The number of communication rounds needed to reach a certain test accuracy in the experiments on FMNIST, CIFAR10 and CIFAR100. The columns "extra comp." and "extra comm." denote the computation and communication complexity of additional operations in each sampling scheme compared to random sampling. All results are for the concentration parameter setting (2).

| | FMNIST | | CIFAR10 | | CIFAR100 | | extra | extra |
|---|---|---|---|---|---|---|---|---|
| Scheme | acc = 0.75 | speedup | acc = 0.6 | speedup | acc = 0.3 | speedup | comp. | comm. |
| Random | 149 | $1.0\times$ | 898 | $1.0\times$ | 549 | $1.0\times$ | - | - |
| pow-d | 79 | $1.8\times$ | 1037 | $0.9\times$ | 770 | $0.71\times$ | $\mathcal{O}(|\theta^t|)$ | $\mathcal{O}(|\theta^t|)$ |
| CS | 114 | $1.3\times$ | 748 | $1.2\times$ | 530 | $1.03\times$ | $\mathcal{O}(|\theta^t|)$ | - |
| DivFL | 478 | $0.3\times$ | 1417 | $0.6\times$ | 1345 | $0.4\times$ | $\mathcal{O}(|\theta^t|)$ | $\mathcal{O}(|\theta^t|)$ |
| **HiCS-FL** | **60** | **$2.5\times$** | **123** | **$7.3\times$** | **336** | **$1.63\times$** | $\mathcal{O}(C)$ | - |

## 5 CONCLUSION

In this paper, we studied federated learning systems where clients that own non-IID data collaboratively train a global model; the system operates under communication constraints and thus only a fraction of clients participates in any given round of training. We developed HiCS-FL, a hierarchical clustered sampling method which estimates clients' data heterogeneity and uses this information to cluster and select clients to participate in training. We analyzed the performance of the proposed heterogeneity estimation method, and the convergence of training a FL system that deploys HiCS-FL. Extensive benchmarking experiments on three datasets demonstrated significant benefits of the proposed method, including improvement in convergence speed, variance and test accuracy, accomplished with only a minor computational overhead.

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

# A  APPENDIX

## A.1  DETAILS OF THE EXPERIMENTS

### A.1.1  GENERAL SETTINGS

The experimental results were obtained using Pytorch (Paszke et al., 2019). In the experiments involving FMNIST, each client used a CNN-based classifier with two $5 \times 5$-convolutional layers and two $2 \times 2$-maxpooling layers (with a stride of 2), followed by a fully-connected layer. In the experiments involving CIFAR10/CIFAR100, each client used a CNN-based classifier with three $3 \times 3$-convolutional layers and two $2 \times 2$-maxpooling layers (with a stride of 2), followed by two fully-connected layers; dimension of the hidden layer was $64$. The optimizers used for model training in the experiments on FMNIST and CIFAR10/CIFAR100 were the mini-batch stochastic gradient descent (SGD) and Adam (Kingma & Ba, 2014), respectively. The learning rate was initially set to $0.001$ and then decreased every 10 iterations, with a decay factor $0.5$. The number of global communication rounds was set to 200, 500 and 1000 for the experiments on FMNIST, CIFAR10 and CIFAR100, respectively. In all the experiments, the number of local epochs $R$ was set to 2 and the size of a mini-batch was set to $64$. The sampling rate (fraction of the clients participating in a training round) was set to $0.1$ for the experiments on FMNIST, and to $0.2$ for the experiments on CIFAR10/CIFAR100. For the sake of visualization, data points in the presented graphs were smoothened by a Savitzky–Golay filter (Schafer, 2011) with window length 13 and the polynomial order set to 3.

### A.1.2  HYPER-PARAMETERS

In all experiments, the hyper-parameter $\mu$ of the regularization term in FedProx (Li et al., 2020b) was set to $0.1$. In the Power-of-Choice (pow-d) (Cho et al., 2020) selection strategy, $d$ was set to the total number of clients: $50$ in the experiments on FMNIST and CIFAR10, $100$ in the experiments on CIFAR100. When running DivFL (Balakrishnan et al., 2022), we used the ideal setting where 1-step gradients were requested from all client in each round (regardless of their participation status), similar to the Power-of-Choice settings. For HiCS-FL (our method), the scaling parameter $T$ (temperature) used in data heterogeneity estimation was set to $0.0025$ in the experiments on FMNIST and to $0.0015$ in the experiments on CIFAR10/CIFAR100. In all experiments, parameter $\lambda$ which multiplies the difference between clients' estimated data heterogeneity (used in clustering) was set to $0.1$. In all experiments, the number of clusters $m$ was for convenience set to be equal to the number of selected clients $K$. The coefficient $\gamma^0$ was set to $4$ in the experiments on FMNIST and CIFAR10 while set to $2$ in the experiments on CIFAR100. To group clients, both Clustered Sampling (Fraboni et al., 2021) and HiCS-FL (our method) utilized an off-the-shelf clustering algorithm performing hierarchical clustering with Ward's Method.

## A.2  EMPIRICAL VALIDATION OF ASSUMPTION 1

To illustrate and empirically validate Assumption 1, we conducted extensive experiments on FM-NIST and CIFAR10 with the same model mentioned in Section A.1. In particular, we varied $\alpha$ over 250 values in the interval $[0.01, 50]$ to generate data partitions allocated to 250 clients; entropy of the generated label distributions ranged from 0 to $\ln 10$ (maximum). In these experiments, we allowed all clients to participate in each of 500 training rounds. To facilitate the desired study, in addition to these 250 clients we also simulated a super-client which owns a data set aggregating the data from all the clients (the set of labels in the aggregated dataset is uniformly distributed). In each round, clients start from the initialized global model and compute local gradients on their datasets; the super-client does the same on the aggregated dataset. The server computes and records squared Euclidean norm of the difference between the local gradients and the "true" gradient (i.e., the super-client's gradient). In each round, the difference between the local gradient and the true gradient changes in a pattern similar to what is stated in Assumption 1. As an illustration, we plot all such gradient differences computed during the entire training process of a client. Specifically, the server computes the difference between local gradient and the true gradient in each round of training, obtaining $250 \times 500 = 12500$ data points that correspond to 250 data partitions. For better visualization, we merged adjacent points.

The results obtained by following these steps in experiments on FMNIST and CIFAR10 are shown in Figure 6. For a more informative visualization, the horizontal coordinate of a point in the scatter plot is $H(\mathcal{D}^{(k)})$, while the vertical coordinate is $\|\eta_t \nabla F_k(\theta^t) - \eta_t \nabla F(\theta^t)\|^2$. The dashed lines correspond to the curves $y = -\exp(\beta [x - H(\mathcal{D}_0)])\rho + \kappa$ that envelop the majority of the generated points. In the case of FMNIST, the blue dashed line is parametrized by $\beta = 1.0$, $\rho = 0.13$, and $\kappa = 0.14$ while the green dashed line is parametrized by $\beta = 1.5$, $\rho = 0.025$, and $\kappa = 0.022$; these two lines envelop 95% of the generated points. In the case of CIFAR10, the blue dashed line is parametrized by $\beta = 2.0$, $\rho = 0.30$, and $\kappa = 0.36$ while the green dashed line is parametrized by $\beta = 1.8$, $\rho = 0.15$, and $\kappa = 0.20$; as in the other plot, these two lines envelop 95% of the generated points. As the plots indicate, the difference between the local gradient and the true gradient increases as $H(\mathcal{D}^{(k)})$ decreases, implying that the local gradient computed by a client with more balanced data is closer to the true gradient.

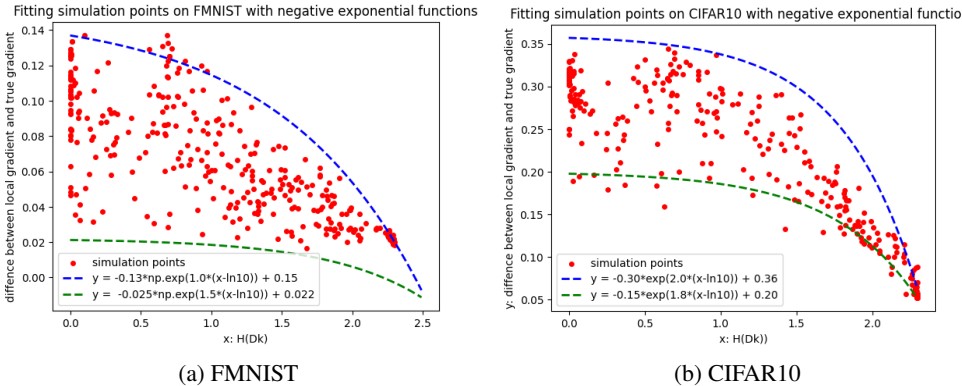

|  |  |
|---|---|
| (a) FMNIST | (b) CIFAR10 |

Figure 6: Visualization of the difference between local gradients and the global gradient (evaluated if all the data is centrally collected).

### A.3 GRADIENT OF THE OUTPUT (FULLY CONNECTED) LAYER'S BIAS

Given a batch of samples $(\mathbf{x}^{(j,n)}, y^{(j,n)})$, the cross-entropy loss is readily computed as

$$\mathcal{L}_{\mathbf{CE}} = -\frac{1}{Bl}\sum_{j=1}^{l}\sum_{n=1}^{B} \log \frac{\exp\left(q_{y^{(j,n)}}^{(j,n)}\right)}{\sum_{c=1}^{C}\exp\left(q_c^{(j,n)}\right)} = \frac{1}{Bl}\sum_{j=1}^{l}\sum_{n=1}^{B}\mathcal{L}_{\mathbf{CE}}^{(j,n)}, \quad y^{(j,n)} \in [C] \quad (13)$$

$$q_c^{(j,n)} = \sum_{d=1}^{L} w_{d,c} z_d^{(j,n)} + b_c, \quad (14)$$

where $B$ is the batchsize; $l$ is the number of mini-batches; $C$ is the number of classes; $d$ is the dimension of the hidden space; $z_d^{(j,n)}$ denotes the $d$-th feature in the hidden space given sample $\mathbf{x}^{(j,n)}$ in the $j$-th batch; $w_{d,c}$ and $b_c$ denote the weight of $z_d^{(j,n)}$ and the bias for the neuron that outputs the probability of the class $c$, respectively; and $q_c^{(j,n)}$ is the corresponding output logit on class $c$. The gradient of the bias $b_i$ given sample $\mathbf{x}^{(j,n)}$ can be computed by the chain rule as

$$\frac{\partial \mathcal{L}_{\mathbf{CE}}^{(j,n)}}{\partial b_i} = -\frac{\partial \mathcal{L}_{\mathbf{CE}}^{(j,n)}}{\partial Q} \cdot \frac{\partial Q}{\partial q_i^{(j,n)}} \cdot \frac{\partial q_i^{(j,n)}}{\partial b_i}, \quad (15)$$

where

$$Q = \frac{\exp\left(q_{y^{(j,n)}}^{(j,n)}\right)}{\sum_{c=1}^{C}\exp\left(q_c^{(j,n)}\right)}. \quad (16)$$

Then

$$\frac{\partial \mathcal{L}_{\mathbf{CE}}^{(j,n)}}{\partial Q} = \frac{1}{Q}, \quad \frac{\partial q_i^{(j,n)}}{\partial b_i} = 1. \quad (17)$$

If $i = y^{(j,n)}$,

$$\frac{\partial Q}{\partial q_i^{(j,n)}} = \frac{\exp\left(q_{y^{(j,n)}}^{(j,n)}\right)\left(\sum_{c=1}^{C}\exp\left(q_c^{(j,n)}\right)\right) - \exp\left(q_{y^{(j,n)}}^{(j,n)}\right)^2}{\left(\sum_{c=1}^{C}\exp\left(q_c^{(j,n)}\right)\right)^2} = \frac{Q\sum_{c \neq y^{(j,n)}}\exp\left(q_c^{(j,n)}\right)}{\sum_{c=1}^{C}\exp\left(q_c^{(j,n)}\right)}.$$

(18)

If $i \neq y^{(j,n)}$,

$$\frac{\partial Q}{\partial q_i^{(j,n)}} = -\frac{\exp\left(q_{y^{(j,n)}}^{(j,n)}\right)\exp\left(q_i^{(j,n)}\right)}{\left(\sum_{c=1}^{C}\exp\left(q_c^{(j,n)}\right)\right)^2} = -\frac{Q\exp\left(q_i^{(j,n)}\right)}{\sum_{c=1}^{C}\exp\left(q_c^{(j,n)}\right)}.$$

(19)

By plugging Eq. 18 and 19 in Eq. 15, we obtain

$$\frac{\partial \mathcal{L}_{\mathbf{CE}}^{(j,n)}}{\partial b_i} = -\frac{\sum_{c \neq y^{(j,n)}}\exp\left(q_c^{(j,n)}\right)}{\sum_{c=1}^{C}\exp\left(q_c^{(j,n)}\right)}, \text{ if } i = y^{(j,n)}; \frac{\partial \mathcal{L}_{\mathbf{CE}}^{(j,n)}}{\partial b_i} = \frac{\exp\left(q_i^{(j,n)}\right)}{\sum_{c=1}^{C}\exp\left(q_c^{(j,n)}\right)}, \text{ if } i \neq y^{(j,n)}.$$

(20)

## A.4 EXPECTATION OF THE LOCAL UPDATE $\Delta\mathbf{b}^{(k)}$

By combining Eq. 4 and 5 and taking expectation, we obtain

$$
\begin{aligned}
\mathbb{E}\left[\Delta b_i^{(k)}\right] &= -\frac{\eta}{Bl}\sum_{j=1}^{l}\sum_{n=1}^{B}\sum_{r=1}^{R}\mathbb{E}\left[\nabla_{b_i}\mathcal{L}_{\mathbf{CE}}^{(j,n,r)}\right] \\
&= \eta\sum_{r=1}^{R}\mathbb{P}\{i = y^{(j,n)}\}\mathbb{I}\{i = y^{(j,n)}\}\frac{1}{Bl}\sum_{j=1}^{l}\sum_{n=1}^{B}\frac{\sum_{c \neq i}\exp(q_c^{(j,n,r)})}{\sum_{c=1}^{C}\exp(q_c^{(j,n,r)})} \\
&\quad -\eta\sum_{r=1}^{R}\mathbb{P}\{i \neq y^{(j,n)}\}\mathbb{I}\{i \neq y^{(j,n)}\}\frac{1}{Bl}\sum_{j=1}^{l}\sum_{n=1}^{B}\frac{\exp(q_i^{(j,n,r)})}{\sum_{c=1}^{C}\exp(q_c^{(j,n,r)})} \\
&= \eta\sum_{r=1}^{R}D_i^{(k)}\sum_{c \neq i}\mathbb{E}_{(\mathbf{x},y)\sim\mathcal{B}^{-c}}\left[\frac{\exp(q_c^{(j,n,r)})}{\sum_{c=1}^{C}\exp(q_c^{(j,n,r)})}\right] \\
&\quad -\eta\sum_{r=1}^{R}(1 - D_i^{(k)})\mathbb{E}_{(\mathbf{x},y)\sim\mathcal{B}^{-i}}\left[\frac{\exp(q_i^{(j,n,r)})}{\sum_{c=1}^{C}\exp(q_c^{(j,n,r)})}\right] \\
&= \eta\sum_{r=1}^{R}D_i^{(k)}\sum_{c \neq i}\mathbb{E}_{(\mathbf{x},y)\sim\mathcal{B}^{-c}}\left[\mathbf{s}_c^{-c}(\mathbf{x})\right] - \eta\sum_{r=1}^{R}(1 - D_i^{(k)})\mathbb{E}_{(\mathbf{x},y)\sim\mathcal{B}^{-i}}\left[\mathbf{s}_i^{-i}(\mathbf{x})\right] \\
&= \eta R\left(D_i^{(k)}\sum_{c \neq i}\mathcal{E}_c - (1 - D_i^{(k)})\mathcal{E}_i\right) \\
&= \eta R\left(D_i^{(k)}\sum_{c=1}^{C}\mathcal{E}_c - \mathcal{E}_i\right).
\end{aligned}
$$

(21)

Note that

$$
\begin{aligned}
\sum_{i=1}^{C} \mathcal{E}_i &= \sum_{i=1}^{C} \mathbb{E}_{(\mathbf{x},y) \sim \mathcal{B}^{-i}} \left[ \mathbf{s}_i^{-i}(\mathbf{x}) \right] \\
&= \mathbb{E} \left[ \sum_{i=1}^{C} \frac{1}{C-1} \sum_{c \neq i} \frac{1}{BlD_c^{(k)}} \sum_{j=1}^{l} \sum_{n=1}^{B} \mathbb{I}\{y^{(j,n)} = c\} \frac{\exp(q_i^{(j,n)})}{\sum_{c=1}^{C} \exp(q_c^{(j,n)})} \right] \\
&= \frac{1}{C-1} \sum_{i=1}^{C} \frac{1}{BlD_i^{(k)}} \sum_{j=1}^{l} \sum_{n=1}^{B} \mathbb{P}\{y^{(j,n)} = i\} \frac{\sum_{c \neq i} \exp(q_c^{(j,n)})}{\sum_{c=1}^{C} \exp(q_c^{(j,n)})} \\
&= -\frac{C}{C-1} \frac{1}{Bl} \sum_{j=1}^{l} \sum_{n=1}^{B} \frac{\exp(q_{y^{(j,n)}}^{(j,n)})}{\sum_{c=1}^{C} \exp(q_c^{(j,n)})} + \frac{C}{C-1}
\end{aligned}
\tag{22}
$$

A comparison to $\mathcal{L}_{\mathbf{CE}}$ in Eq. 13 reveals that as $\mathcal{L}_{\mathbf{CE}}$ decreases during training, so does $\sum_{i=1}^{C} \mathcal{E}_i$. Given an untrained/initialized neural network model, $\mathcal{E}_i^0 = 1/C$ for $\forall i \in [C]$, i.e., $\sum_{i=1}^{C} \mathcal{E}_i^0 = -\frac{1}{C-1} + \frac{C}{C-1} = 1$. At global round $T$, if $\mathcal{L}_{\mathbf{CE}}^* = 0$, then $\sum_{i=1}^{C} \mathcal{E}_i^T = -\frac{C}{C-1} + \frac{C}{C-1} = 0$.

## A.5 Privacy of $\mathcal{D}^{(k)}$

According to Eq. 6, the server is able to obtain $C$ linear equations from each client,

$$
\mathbb{E}\left[\Delta b_i^{(k)}\right] = \eta R \left( D_i^{(k)} \sum_{c=1}^{C} \mathcal{E}_c - \mathcal{E}_i \right), \text{ for } \forall i \in [C],
\tag{23}
$$

$$
\sum_{i=1}^{C} D_i^{(k)} = 1,
\tag{24}
$$

where $C$ denotes the number of classes. Suppose $\mathbb{E}[\Delta b_i^{(k)}]$ are known by the server. Then $D_i^{(k)}$, the variables in the aforementioned equations, cannot be determined uniquely since there are $C$ variables and $C + 1$ equations. Therefore, the server is unable to infer clients' true data label distribution and the privacy of $\mathcal{D}^{(k)}$ is protected.

## A.6 Proof of Theorem 1

In Section A.3 we derived an expression for the gradient of the bias in the output layer given a single sample $(\mathbf{x}^{(j,n)}, y)$ in the mini-batch. It is worthwhile making the following two observations:

- the sign of the $y^{(j,n)}$-th component of $\nabla_{\mathbf{b}} \mathcal{L}_{\mathbf{CE}}^{(j,n)}(\mathbf{x}^{(j,n)}, y^{(j,n)})$ is opposite of the sign of the other components; and

- the $y^{(j,n)}$-th component of $\nabla_{\mathbf{b}} \mathcal{L}_{\mathbf{CE}}^{(j,n)}(\mathbf{x}^{(j,n)}, y^{(j,n)})$ is equal in magnitude to all other components combined.

*Proof:* Let $\Delta \mathbf{b}^{(k)} = [\Delta b_1^{(k)}, \dots, \Delta b_C^{(k)}]$ denote the local update (made by client $k$) of the bias in the output layer of the neural network model, and let $\mathcal{D}^{(k)} = [D_1^{(k)}, \dots, D_C^{(k)}]$ be the (unknown) true data label distribution, $\sum_{i=1}^{C} D_i^{(k)} = 1$. Assuming the learning rate $\eta$ and $R$ local epochs, the expectation of the local update of $\Delta \mathbf{b}^{(k)}$ is

$$
\mathbb{E}\left[\Delta b_i^{(k)}\right] = \eta R \left( D_i^{(k)} \sum_{c=1}^{C} \mathcal{E}_c - \mathcal{E}_i \right).
\tag{25}
$$

Data heterogeneity can be captured via entropy, $H(\mathcal{D}^{(k)}) = -\sum_{c=1}^{C} D_i^{(k)} \ln D_i^{(k)}$, where higher $H(\mathcal{D}^{(k)})$ indicates that client $k$ has more balanced data. However, since we do not have access to

the client's data distribution, we instead define and use as a measure of heterogeneity $\hat{H}(\mathcal{D}^{(k)}) \triangleq H(\text{softmax}(\Delta\mathbf{b}^{(k)}, T))$, where

$$\text{softmax}(\Delta\mathbf{b}^{(k)}, T)_i = \frac{\exp(\Delta b_i^{(k)}/T)}{\sum_{c=1}^{C} \exp(\Delta b_c^{(k)}/T)}, \tag{26}$$

and where $T$ denotes the *temperature* of the softmax operator. Suppose there are two clients, $u$ and $k$, with class-balanced and class-imbalanced data; let $\mathcal{D}^{(u)}$ and $\mathcal{D}^{(k)}$ denote their data label distributions, respectively, while $\hat{\mathcal{D}}^{(u)}$ and $\hat{\mathcal{D}}^{(k)}$ are computed by $\text{softmax}(\Delta\mathbf{b}^{(u)}, T)$ and $\text{softmax}(\Delta\mathbf{b}^{(k)}, T)$. Without a loss of generality, we can re-parameterize $\hat{\mathcal{D}}^{(u)}$ as

$$\hat{\mathcal{D}}^{(u)} = \epsilon\mathbf{U} + \sum_{i=1}^{C} \epsilon_i\mathbf{Z}_i, \tag{27}$$

where $\mathbf{U} = [\frac{1}{C}, \dots, \frac{1}{C}]$ denotes uniform distribution; $i$-th component of $\mathbf{Z}_i$ is 1 while the remaining components are 0; $\epsilon$ and $\epsilon_i$ are all non-negative such that $\epsilon + \sum_{i=1}^{C} \epsilon_i = 1$. We can always set $\min_j \epsilon_j = 0$; otherwise, let $\epsilon' = \epsilon + \min_j \epsilon_j$ and $\epsilon_i' = \epsilon_i - \min_j \epsilon_j$, $\forall i \in [C]$; $\epsilon$ quantifies how close is $\hat{\mathcal{D}}^{(u)}$ to $\mathbf{U}$. Due to the concavity of entropy,

$$H(\hat{\mathcal{D}}^{(u)}) \geq \epsilon H(\mathbf{U}) + \sum_{i=1}^{C} \epsilon_i H(\mathbf{Z}_i) = \epsilon \ln C. \tag{28}$$

We will find the following lemma useful.

**Lemma 1** *For two probability vectors $\boldsymbol{p}$ and $\boldsymbol{q}$ with dimension $C$, the Kullback–Leibler divergence between $\boldsymbol{p}$ and $\boldsymbol{q}$ satisfies*

$$KLD(\boldsymbol{p}||\boldsymbol{q}) \geq \frac{1}{2}\|\mathbf{p} - \mathbf{q}\|_1^2, \tag{29}$$

*where $\|\mathbf{p} - \mathbf{q}\|_1 = \sum_{i=1}^{C} |p_i - q_i|$.*

For the proof of the lemma, please see (Dragomir et al., 2000). Applying it, we obtain

$$\mathbf{KLD}(\hat{\mathcal{D}}^{(k)}||\mathbf{U}) = H(\mathbf{U}) - H(\hat{\mathcal{D}}^{(k)}) \geq \frac{1}{2}\left\|\hat{\mathcal{D}}^{(k)} - \mathbf{U}\right\|_1^2 \geq \frac{1}{2}\left\|\hat{\mathcal{D}}^{(k)} - \mathbf{U}\right\|_2^2. \tag{30}$$

Combining Eq. 28 and Eq. 30, we obtain

$$H(\hat{\mathcal{D}}^{(u)}) - H(\hat{\mathcal{D}}^{(k)}) \geq (\epsilon - 1) \ln C + \frac{1}{2}\left\|\hat{\mathcal{D}}^{(k)} - \mathbf{U}\right\|_2^2. \tag{31}$$

By taking expectations of both sides,

$$\mathbb{E}\left[H(\hat{\mathcal{D}}^{(u)}) - H(\hat{\mathcal{D}}^{(k)})\right] \geq (\mathbb{E}[\epsilon] - 1) \ln C + \frac{1}{2}\mathbb{E}\left[\left\|\hat{\mathcal{D}}^{(k)} - \mathbf{U}\right\|_2^2\right]. \tag{32}$$

Since $\left\|\hat{\mathcal{D}}^{(k)} - \mathbf{U}\right\|_2^2$ is convex (composition of the Euclidean norm and softmax), according to Jensen's inequality

$$\mathbb{E}\left[H(\hat{\mathcal{D}}^{(u)}) - H(\hat{\mathcal{D}}^{(k)})\right] \geq (\mathbb{E}[\epsilon] - 1) \ln C + \frac{1}{2}\left\|\hat{\mathcal{D}}^{(k)}(\mathbb{E}[\Delta\mathbf{b}^{(k)}]) - \mathbf{U}\right\|_2^2, \tag{33}$$

where

$$\hat{\mathcal{D}}^{(k)}(\mathbb{E}[\Delta\mathbf{b}^{(k)}])_i = \frac{\exp\left(\eta R\left(D_i^{(k)} \sum_{c=1}^{C} \mathcal{E}_c - \mathcal{E}_i\right)/T\right)}{\sum_j^C \exp\left(\eta R\left(D_j^{(k)} \sum_{c=1}^{C} \mathcal{E}_c - \mathcal{E}_j\right)/T\right)}. \tag{34}$$

Selecting $T$ such that $\eta R\left(D_i^{(k)} \sum_{c=1}^{C} \mathcal{E}_c - \mathcal{E}_i\right)/T$ is sufficiently small and applying the first-order Taylor's expansion of $e^x$ around 0, we obtain

$$\sum_j^C \exp\left(\eta R\left(D_j^{(k)} \sum_{c=1}^{C} \mathcal{E}_c - \mathcal{E}_j\right)/T\right) = \sum_j^C 1 + \eta R \sum_j^C \left(D_j^{(k)} \sum_{c=1}^{C} \mathcal{E}_c - \mathcal{E}_j\right)/T = C, \tag{35}$$

where $\sum_{j=1}^{C} D_j^{(k)} = 1$. This leads to a simplified $\hat{\mathcal{D}}^{(k)}(\mathbb{E}[\Delta \mathbf{b}^{(k)}])$,

$$\hat{\mathcal{D}}^{(k)}(\mathbb{E}[\Delta \mathbf{b}^{(k)}])_i = \frac{1 + \eta R \left( D_i^{(k)} \sum_{c=1}^{C} \mathcal{E}_c - \mathcal{E}_i \right)/T}{C}. \tag{36}$$

Substituting Eq. 36 for the second term on the right-hand side of ineq. 33 leads to

$$\left\| \hat{\mathcal{D}}^{(k)}(\mathbb{E}[\Delta \mathbf{b}^{(k)}]) - \mathbf{U} \right\|_2^2 = \left( \frac{\eta R}{CT} \right)^2 \sum_{i=1}^{C} \left( D_i^{(k)} \sum_{c=1}^{C} \mathcal{E}_c - \mathcal{E}_i \right)^2. \tag{37}$$

Now, consider

$$\hat{\mathcal{D}}^{(u)} - \mathbf{U} = (\epsilon - 1)\mathbf{U} + \sum_{i=1}^{C} \epsilon_i \mathbf{Z}_i. \tag{38}$$

Taking expectations of both sides,

$$\mathbb{E}\left[ (\epsilon - 1)\mathbf{U} + \sum_{i=1}^{C} \epsilon_i \mathbf{Z}_i \right] = \mathbb{E}\left[ \hat{\mathcal{D}}^{(u)} - \mathbf{U} \right] \geq \hat{\mathcal{D}}^{(u)}(\mathbb{E}[\Delta \mathbf{b}^{(u)}]) - \mathbf{U}. \tag{39}$$

The above inequality holds component-wise, so for the $j$-component ($\epsilon_j = 0$)

$$\mathbb{E}[\frac{1}{C}(\epsilon - 1) + \epsilon_j] = \mathbb{E}[\frac{1}{C}(\epsilon - 1)] \geq \hat{\mathcal{D}}^{(u)}(\mathbb{E}[\Delta \mathbf{b}^{(u)}])_j - \mathbf{U}_i = \frac{\eta R \left( D_j^{(u)} \sum_{c=1}^{C} \mathcal{E}_c - \mathcal{E}_j \right)}{CT}. \tag{40}$$

Therefore,

$$\mathbb{E}[\epsilon] - 1 \geq \frac{\eta R \left( D_j^{(u)} \sum_{c=1}^{C} \mathcal{E}_c - \mathcal{E}_j \right)}{T} \geq \min_i \frac{\eta R \left( D_i^{(u)} \sum_{c=1}^{C} \mathcal{E}_c - \mathcal{E}_i \right)}{T}. \tag{41}$$

Taking absolute value of both sides yields

$$|\mathbb{E}[\epsilon] - 1| \leq \frac{\eta R}{T} \max_i \left| D_i^{(u)} \sum_{c=1}^{C} \mathcal{E}_c - \mathcal{E}_i \right| = \frac{\eta R}{T} \max_i \left| (D_i^{(u)} - \frac{1}{C}) \sum_{c=1}^{C} \mathcal{E}_c - \mathcal{E}_i + \frac{1}{C} \sum_{c=1}^{C} \mathcal{E}_c \right|. \tag{42}$$

By applying the triangle inequality we obtain

$$|\mathbb{E}[\epsilon] - 1| \leq \frac{\eta R}{T} \max_i \left| D_i^{(u)} - \frac{1}{C} \right| \sum_{c=1}^{C} \mathcal{E}_c + \frac{\eta R}{T} \max_i \left| \frac{1}{C} \sum_{c=1}^{C} \mathcal{E}_c - \mathcal{E}_i \right|. \tag{43}$$

Let $\delta = \max_i \left| \frac{1}{C} \sum_{c=1}^{C} \mathcal{E}_c - \mathcal{E}_i \right|$. Since $\sum_{c=1}^{C} \mathcal{E}_c \leq C\frac{1}{C} = 1$, it holds that

$$|\mathbb{E}[\epsilon] - 1| \leq \frac{\eta R}{T} \max_i \left| D_i^{(u)} - \frac{1}{C} \right| + \frac{\eta R}{T} \delta. \tag{44}$$

Furthermore, since $\mathbb{E}[\epsilon] - 1 < 0$,

$$\mathbb{E}[\epsilon] - 1 \geq -\frac{\eta R}{T} \max_i \left| D_i^{(u)} - \frac{1}{C} \right| - \frac{\eta R}{T} \delta. \tag{45}$$

Note that

$$
\begin{aligned}
\left( D_i^{(k)} \sum_{c=1}^{C} \mathcal{E}_c - \mathcal{E}_i \right)^2 &= \left( (D_i^{(k)} - \frac{1}{C}) \sum_{c=1}^{C} \mathcal{E}_c - \mathcal{E}_i + \frac{1}{C} \sum_{c=1}^{C} \mathcal{E}_c \right)^2 \\
&= \left( (D_i^{(k)} - \frac{1}{C}) \sum_{c=1}^{C} \mathcal{E}_c \right)^2 + \left( \frac{1}{C} \sum_{c=1}^{C} \mathcal{E}_c - \mathcal{E}_i \right)^2 \\
&\quad + 2 \left( \sum_{c=1}^{C} \mathcal{E}_c \right) \left( D_i^{(k)} - \frac{1}{C} \right) \left( \frac{1}{C} \sum_{c=1}^{C} \mathcal{E}_c - \mathcal{E}_i \right) \\
&\geq \left( (D_i^{(k)} - \frac{1}{C}) \sum_{c=1}^{C} \mathcal{E}_c \right)^2 + 2 \left( \sum_{c=1}^{C} \mathcal{E}_c \right) \left( D_i^{(k)} - \frac{1}{C} \right) \left( \frac{1}{C} \sum_{c=1}^{C} \mathcal{E}_c - \mathcal{E}_i \right).
\end{aligned}
\tag{46}
$$

Therefore,

$$
\sum_{i=1}^{C} \left( D_i^{(k)} \sum_{c=1}^{C} \mathcal{E}_c - \mathcal{E}_i \right)^2 \geq \left( \sum_{c=1}^{C} \mathcal{E}_c \right)^2 \sum_{i=1}^{C} \left( D_i^{(k)} - \frac{1}{C} \right)^2
$$

$$
+ 2 \left( \sum_{c=1}^{C} \mathcal{E}_c \right) \sum_{i=1}^{C} \left( D_i^{(k)} - \frac{1}{C} \right) \left( \frac{1}{C} \sum_{c=1}^{C} \mathcal{E}_c - \mathcal{E}_i \right)
$$

$$
= \left( \sum_{c=1}^{C} \mathcal{E}_c \right)^2 \sum_{i=1}^{C} \left( D_i^{(k)} - \frac{1}{C} \right)^2
$$

$$
+ 2 \left( \sum_{c=1}^{C} \mathcal{E}_c \right) \sum_{i=1}^{C} \left( \frac{D_i^{(k)}}{C} \sum_{c=1}^{C} \mathcal{E}_c - \frac{1}{C^2} \sum_{c=1}^{C} \mathcal{E}_c + \frac{\mathcal{E}_i}{C} - D_i^{(k)} \mathcal{E}_i \right)
$$

$$
= \left( \sum_{c=1}^{C} \mathcal{E}_c \right)^2 \sum_{i=1}^{C} \left( D_i^{(k)} - \frac{1}{C} \right)^2
$$

$$
+ 2 \left( \sum_{c=1}^{C} \mathcal{E}_c \right) \left( \frac{1}{C} \sum_{c=1}^{C} \mathcal{E}_c - \frac{1}{C} \sum_{c=1}^{C} \mathcal{E}_c + \frac{1}{C} \sum_{i=1}^{C} \mathcal{E}_i - \sum_{i=1}^{C} D_i^{(k)} \mathcal{E}_i \right)
$$

$$
\geq \left( \sum_{c=1}^{C} \mathcal{E}_c \right)^2 \sum_{i=1}^{C} \left( D_i^{(k)} - \frac{1}{C} \right)^2 + 2 \left( \sum_{c=1}^{C} \mathcal{E}_c \right) \left( \frac{1}{C} \sum_{c=1}^{C} \mathcal{E}_c - \max_j \mathcal{E}_j \right)
$$

$$
\geq \left( \sum_{c=1}^{C} \mathcal{E}_c \right)^2 \sum_{i=1}^{C} \left( D_i^{(k)} - \frac{1}{C} \right)^2 - 2\delta.
$$

$$(47)$$

Substituting the above expression in Eq. 33, we obtain

$$
\mathbb{E}\left[ H(\hat{\mathcal{D}}^{(u)}) - H(\hat{\mathcal{D}}^{(k)}) \right] \geq -\frac{\eta R \ln C}{T} \max_j \left| D_j^{(u)} - \frac{1}{C} \right| - \frac{\eta R \ln C}{T} \delta \tag{48}
$$

$$
+ \frac{1}{2} \left( \frac{\eta R}{CT} \right)^2 \left( \sum_{c=1}^{C} \mathcal{E}_c \right)^2 \sum_{i=1}^{C} \left( D_i^{(k)} - \frac{1}{C} \right)^2 - \left( \frac{\eta R}{CT} \right)^2 \delta, \tag{49}
$$

and, therefore,

$$
\mathbb{E}\left[ H(\hat{\mathcal{D}}^{(u)}) - H(\hat{\mathcal{D}}^{(k)}) \right] \geq \frac{1}{2} \left( \frac{\eta R}{CT} \sum_{c=1}^{C} \mathcal{E}_c \right)^2 \left\| \mathcal{D}^{(k)} - \mathbf{U} \right\|_2^2 - \frac{\eta R \ln C}{T} \left\| \mathcal{D}^{(u)} - \mathbf{U} \right\|_\infty - \mathcal{C}\delta, \tag{50}
$$

where $\mathcal{C} = \frac{\eta R(\eta R + C^2 T \ln C)}{C^2 T^2}$. ∎

## A.7 CONVERGENCE ANALYSIS

Here we present the convergence analysis of an FL system deploying FedAvg with SGD wherein only a small fraction of clients participates in any given round of training. Recall that the objective function that comes up when training a neural network model is generally non-convex; we make the standard assumptions of smoothness, unbiased gradient estimate, and bounded variance.

**Assumption 2 (Smoothness)** *Each local objective function $F_k(\cdot)$ is L-smooth,*

$$
\left\| \nabla F_k(\theta_k^{t+1}) - \nabla F_k(\theta_k^t) \right\|_2 \leq L \left\| \theta_k^{t+1} - \theta_k^t \right\|_2. \tag{51}
$$

**Assumption 3 (Gradient oracle)** *The stochastic gradient estimator $g_k(\theta_k^{t,r}) = \nabla F_k(\theta_k^{t,r}) + \zeta_k^{t,r}$ for each global round $t$ and local epoch $r$ is such that*

$$
\mathbb{E}[\zeta_k^{t,r}] = 0 \tag{52}
$$

*and*

$$\mathbb{E}\left[\left\|\zeta_k^{t,r}\right\|^2 |\theta_k^{t,r}\right] \le \sigma^2. \tag{53}$$

With these three assumptions in place, we provide the proof of Theorem 2 stated in the main paper. The proof relies on the technique previously used in (Chen et al., 2020; Yang et al., 2021a), where the sampling method is unbiased and thus $\mathbb{E}\left[\frac{1}{K}\sum_{k\in\mathcal{S}^t}\sum_{r=1}^R g_k(\theta_k^{t,r})\right] = \sum_{k=1}^N\sum_{r=1}^R p_k\nabla F_k(\theta_k^{t,r})$. We provide a generalization that holds for any sampling strategy, resulting in $\mathbb{E}\left[\frac{1}{K}\sum_{k\in\mathcal{S}^t}\sum_{r=1}^R g_k(\theta_k^{t,r})\right] = \sum_{k=1}^N\sum_{r=1}^R \omega_k^t\nabla F_k(\theta_k^{t,r})$, where $\omega_k^t$ denotes the probability of sampling client $k$ in round $t$ under sampling strategy $\Pi$. Note that $\sum_{k=1}^N\omega_k^t = 1$. We assume that all clients deploy the same number of local epochs $R$ and use learning rate $\eta$ at round $t$.

### A.7.1 KEY LEMMA

**Lemma 2** *(Lemma 2 in (Yang et al., 2021a)) Instate Assumptions 1, 2 and 3. For any step size $\eta$ such that $\eta \le \frac{1}{8LR}$, for any client $k$ it holds that*

$$\mathbb{E}\left[\left\|\theta_k^{t,r} - \theta^t\right\|^2\right] \le 5R\eta^2(\sigma^2 + 6R\sigma_k^2) + 30R^2\eta^2\left\|\nabla F(\theta^t)\right\|^2. \tag{54}$$

*Proof of Lemma 2:* For any client $k \in [N]$ and $r \in [R]$,

$$
\begin{aligned}
\mathbb{E}\left[\left\|\theta_k^{t,r} - \theta^t\right\|^2\right] &= \mathbb{E}\left[\left\|\theta_k^{t,r-1} - \theta^t - \eta g_k(\theta_k^{t,r-1})\right\|^2\right] \\
&= \mathbb{E}[\|\theta_k^{t,r-1} - \theta^t - \eta(g_k(\theta_k^{t,r-1}) - \nabla F_k(\theta_k^{t,r-1}) + \nabla F_k(\theta_k^{t,r-1}) \\
&\quad - \nabla F_k(\theta^t) + \nabla F_k(\theta^t) - \nabla F(\theta^t) + \nabla F(\theta^t))\|^2] \\
&\le \left(1 + \frac{1}{2R-1}\right)\mathbb{E}\left\|\theta_k^{t,r-1} - \theta^t\right\|^2 + \eta^2\mathbb{E}\left\|g_k(\theta_k^{t,r-1}) - \nabla F_k(\theta_k^{t,r-1})\right\|^2 \\
&\quad + 6R\eta^2\mathbb{E}\left\|\nabla F_k(\theta_k^{t,r-1}) - \nabla F_k(\theta^t)\right\|^2 + 6R\eta^2\mathbb{E}\left\|g_k(\nabla F_k(\theta^t) - \nabla F(\theta^t)\right\|^2 \\
&\quad + 6R\eta^2\mathbb{E}\left\|\nabla F(\theta^t)\right\|^2 \\
&\le \left(1 + \frac{1}{2R-1}\right)\mathbb{E}\left\|\theta_k^{t,r-1} - \theta^t\right\|^2 + \eta^2\sigma^2 + 6R\eta^2 L^2\mathbb{E}\left\|\theta_k^{t,r-1} - \theta^t\right\|^2 \\
&\quad + 6R\eta^2\sigma_k^2 + 6R\eta^2\mathbb{E}\left\|\nabla F(\theta^t)\right\|^2 \\
&= \left(1 + \frac{1}{2R-1} + 6R\eta^2 L^2\right)\mathbb{E}\left\|\theta_k^{t,r-1} - \theta^t\right\|^2 + \eta^2\sigma^2 + 6R\eta^2\sigma_k^2 \\
&\quad + 6R\eta^2\mathbb{E}\left\|\nabla F(\theta^t)\right\|^2 \\
&\le \left(1 + \frac{1}{R-1}\right)\mathbb{E}\left\|\theta_k^{t,r-1} - \theta^t\right\|^2 + \eta^2\sigma^2 + 6R\eta^2\sigma_k^2 + 6R\eta^2\mathbb{E}\left\|\nabla F(\theta^t)\right\|^2.
\end{aligned}
\tag{55}
$$

Unrolling the recursion yields

$$
\begin{aligned}
\mathbb{E}\left[\left\|\theta_k^{t,r} - \theta^t\right\|^2\right] &\le \sum_{r=1}^R\left(1 + \frac{1}{R-1}\right)^{r-1}\left(\eta^2\sigma^2 + 6R\eta^2\sigma_k^2 + 6R\eta^2\mathbb{E}\left\|\nabla F(\theta^t)\right\|^2\right) \\
&\le (R-1)\left[\left(1 + \frac{1}{R-1}\right)^R - 1\right]\left(\eta^2\sigma^2 + 6R\eta^2\sigma_k^2 + 6R\eta^2\mathbb{E}\left\|\nabla F(\theta^t)\right\|^2\right) \\
&\le 5R\eta^2\left(\sigma^2 + 6R\sigma_k^2\right) + 30R^2\eta^2\left\|\nabla F(\theta^t)\right\|^2.
\end{aligned}
\tag{56}
$$

■

### A.7.2 PROOF OF THEOREM 2

The model update at global round $t$ is formed as

$$\theta^{t+1} = \theta^t - \eta\frac{1}{K}\sum_{k\in\mathbf{S}^t}\sum_{r=1}^{R}g_k(\theta_k^{t,r}), \tag{57}$$

where $\theta^{t+1}$ and $\theta^t$ denote parameters of the global model at rounds $t+1$ and $t$, respectively, and $\theta_k^{t,r}$ denotes parameters of the local model of client $k$ after $r$ local training epochs. Let

$$\Delta^t \triangleq \frac{1}{K}\sum_{k\in\mathbf{S}^t}\sum_{r=1}^{R}g_k(\theta_k^{t,r}). \tag{58}$$

Taking the expectations (conditioned on $\theta^t$) of both sides, we obtain

$$\begin{aligned}
\mathbb{E}\left[F(\theta^{t+1})\right] &= \mathbb{E}\left[F(\theta^t - \eta\Delta^t)\right] \\
&\stackrel{(a)}{\leq} F(\theta^t) - \eta\left\langle\nabla F(\theta^t), \mathbb{E}\left[\Delta^t\right]\right\rangle + \frac{L}{2}\eta^2\mathbb{E}\left[\|\Delta^t\|^2\right] \\
&= F(\theta^t) + \eta\left\langle\nabla F(\theta^t), \mathbb{E}\left[R\nabla F(\theta^t) - R\nabla F(\theta^t) - \Delta^t\right]\right\rangle + \frac{L}{2}\eta^2\mathbb{E}\left[\|\Delta^t\|^2\right] \\
&= F(\theta^t) - R\eta\|\nabla F(\theta^t)\|^2 + \eta\underbrace{\left\langle\nabla F(\theta^t), \mathbb{E}\left[R\nabla F(\theta^t) - \Delta^t\right]\right\rangle}_{A_1} + \frac{L}{2}\eta^2\underbrace{\mathbb{E}\left[\|\Delta^t\|^2\right]}_{A_2}.
\end{aligned} \tag{59}$$

Inequality (a) in the expression above holds due to the smoothness of $F(\cdot)$ (see Assumption 2). Note that the term $A_1$ can be bounded as

$$\begin{aligned}
A_1 &= \left\langle\nabla F(\theta^t), \mathbb{E}\left[R\nabla F(\theta^t) - \Delta^t\right]\right\rangle \\
&= \left\langle\nabla F(\theta^t), \mathbb{E}\left[R\nabla F(\theta^t) - \frac{1}{K}\sum_{k\in\mathcal{S}^t}\sum_{r=1}^{R}g_k(\theta_k^{t,r})\right]\right\rangle \\
&= \left\langle\nabla F(\theta^t), \mathbb{E}\left[R\nabla F(\theta^t)\right] - \sum_{k=1}^{N}\sum_{r=1}^{R}\omega_k^t\nabla F_k(\theta_k^{t,r})\right\rangle \\
&= \sum_{k=1}^{N}\omega_k^t\left\langle\sqrt{R}\nabla F(\theta^t), -\frac{1}{\sqrt{R}}\mathbb{E}\left[\sum_{r=1}^{R}\left(\nabla F_k(\theta_k^{t,r}) - \nabla F(\theta^t)\right)\right]\right\rangle \\
&\stackrel{(a)}{=} \frac{R}{2}\|\nabla F(\theta^t)\|^2 + \frac{1}{2R}\sum_{k=1}^{N}\omega_k^t\mathbb{E}\left\|\sum_{r=1}^{R}\left(\nabla F_k(\theta_k^{t,r}) - \nabla F(\theta^t)\right)\right\|^2 - \frac{1}{2R}\sum_{k=1}^{N}\omega_k^t\mathbb{E}\left\|\sum_{r=1}^{R}\nabla F_k(\theta_k^{t,r})\right\|^2 \\
&\stackrel{(b)}{\leq} \frac{R}{2}\|\nabla F(\theta^t)\|^2 + \frac{1}{R}\sum_{k=1}^{N}\omega_k^t\mathbb{E}\left\|\sum_{r=1}^{R}\left(\nabla F_k(\theta_k^{t,r}) - \nabla F_k(\theta^t)\right)\right\|^2 \\
&\quad + \frac{1}{R}\sum_{k=1}^{N}\omega_k^t\mathbb{E}\left\|\sum_{r=1}^{R}\left(\nabla F_k(\theta^t) - \nabla F(\theta^t)\right)\right\|^2 - \frac{1}{2R}\sum_{k=1}^{N}\omega_k^t\mathbb{E}\left\|\sum_{r=1}^{R}\nabla F_k(\theta_k^{t,r})\right\|^2 \\
&\stackrel{(c)}{\leq} \frac{R}{2}\|\nabla F(\theta^t)\|^2 + \sum_{k=1}^{N}\omega_k^t\sum_{r=1}^{R}\mathbb{E}\left\|\nabla F_k(\theta_k^{t,r}) - \nabla F_k(\theta^t)\right\|^2 \\
&\quad + \sum_{k=1}^{N}\omega_k^t\sum_{r=1}^{R}\mathbb{E}\left\|\nabla F_k(\theta^t) - \nabla F(\theta^t)\right\|^2 - \frac{1}{2R}\sum_{k=1}^{N}\omega_k^t\mathbb{E}\left\|\sum_{r=1}^{R}\nabla F_k(\theta_k^{t,r})\right\|^2 \\
&\stackrel{(d)}{\leq} \frac{R}{2}\|\nabla F(\theta^t)\|^2 + L^2\sum_{k=1}^{N}\omega_k^t\sum_{r=1}^{R}\mathbb{E}\left\|\theta_k^{t,r} - \theta^t\right\|^2 + R\sum_{k=1}^{N}\omega_k^t\sigma_k^2 - \frac{1}{2R}\sum_{k=1}^{N}\omega_k^t\mathbb{E}\left\|\sum_{r=1}^{R}\nabla F_k(\theta_k^{t,r})\right\|^2,
\end{aligned} \tag{60}$$

where equality (a) follows from $\langle \mathbf{x}, \mathbf{y} \rangle = \frac{1}{2} \left( \|\mathbf{x}\|^2 + \|\mathbf{y}\|^2 - \|\mathbf{x} - \mathbf{y}\|^2 \right)$, inequality (b) is due to $\|\mathbf{x} + \mathbf{y}\|^2 \leq 2 \|\mathbf{x}\|^2 + 2 \|\mathbf{y}\|^2$, inequality (c) holds because $\|\sum_{i=1}^{n} \mathbf{z}_i\|^2 \leq n \sum_{i=1}^{n} \|\mathbf{z}_i\|^2$, and inequality (d) follows from Assumptions 1 and 2. By selecting $\eta < \frac{1}{8LR}$ and applying Lemma 2 we obtain

$$
\begin{aligned}
A_1 &\leq \frac{R}{2} \left\| \nabla F(\theta^t) \right\|^2 + L^2 \sum_{k=1}^{N} \omega_k^t \sum_{r=1}^{R} \left[ 5R\eta^2(\sigma^2 + 6R\sigma_k^2) + 30R^2\eta^2 \left\| \nabla F(\theta^t) \right\|^2 \right] \\
&\quad + R \sum_{k=1}^{N} \omega_k^t \sigma_k^2 - \frac{1}{2R} \sum_{k=1}^{N} \omega_k^t \mathbb{E} \left\| \sum_{r=1}^{R} \nabla F_k(\theta_k^{t,r}) \right\|^2 \\
&= \left( \frac{R}{2} + 30L^2 R^3 \eta^2 \right) \left\| \nabla F(\theta^t) \right\|^2 + 5L^2 R^2 \eta^2 \sigma^2 + 30L^2 R^3 \eta^2 \sum_{k=1}^{N} \omega_k^t \sigma_k^2 \\
&\quad + R \sum_{k=1}^{N} \omega_k^t \sigma_k^2 - \frac{1}{2R} \sum_{k=1}^{N} \omega_k^t \mathbb{E} \left\| \sum_{r=1}^{R} \nabla F_k(\theta_k^{t,r}) \right\|^2 .
\end{aligned}
\tag{61}
$$

Furthermore,

$$
\begin{aligned}
A_2 &= \mathbb{E} \left[ \left\| \frac{1}{K} \sum_{k \in \mathcal{S}^t} \sum_{r=1}^{R} g_k(\theta_k^{t,r}) \right\|^2 \right] \\
&= \mathbb{E} \left[ \left\| \sum_{k=1}^{N} \frac{\mathbb{I}\{k \in \mathcal{S}^t\}}{K} \sum_{r=1}^{R} g_k(\theta_k^{t,r}) \right\|^2 \right] \\
&= \mathbb{E} \left[ \left\| \sum_{k=1}^{N} \frac{\mathbb{I}\{k \in \mathcal{S}^t\}}{K} \sum_{r=1}^{R} g_k(\theta_k^{t,r}) - \nabla F_k(\theta_k^{t,r}) + \nabla F_k(\theta_k^{t,r}) \right\|^2 \right] \\
&\overset{(a)}{=} \mathbb{E} \left[ \left\| \sum_{k=1}^{N} \frac{\mathbb{I}\{k \in \mathcal{S}^t\}}{K} \sum_{r=1}^{R} g_k(\theta_k^{t,r}) - \nabla F_k(\theta_k^{t,r}) \right\|^2 \right] + \mathbb{E} \left[ \left\| \sum_{k=1}^{N} \frac{\mathbb{I}\{k \in \mathcal{S}^t\}}{K} \sum_{r=1}^{R} \nabla F_k(\theta_k^{t,r}) \right\|^2 \right] \\
&\overset{(b)}{\leq} \mathbb{E} \left[ \sum_{k=1}^{N} \frac{\mathbb{I}\{k \in \mathcal{S}^t\}}{K} \sum_{r=1}^{R} \left\| g_k(\theta_k^{t,r}) - \nabla F_k(\theta_k^{t,r}) \right\|^2 \right] + \mathbb{E} \left[ \sum_{k=1}^{N} \frac{\mathbb{I}\{k \in \mathcal{S}^t\}}{K} \left\| \sum_{r=1}^{R} \nabla F_k(\theta_k^{t,r}) \right\|^2 \right] \\
&\overset{(c)}{\leq} R\sigma^2 + \sum_{k=1}^{N} \omega_k^t \mathbb{E} \left\| \sum_{r=1}^{R} \nabla F_k(\theta_k^{t,r}) \right\|^2 ,
\end{aligned}
\tag{62}
$$

where equation (a) holds because $\mathbb{E} \left[ g_k(\theta_k^{t,r}) - \nabla F_k(\theta_k^{t,r}) \right] = 0$, inequality (b) stems from the Jensen's inequality, and inequality (c) is due to Assumption 3.

Substituting inequalities (61) and (62) into inequality (59) yields

$$
\begin{aligned}
\mathbb{E} \left[ F(\theta^{t+1}) \right] &\leq F(\theta^t) - R\eta \left\| \nabla F(\theta^t) \right\|^2 + \eta \underbrace{\left\langle \nabla F(\theta^t), \mathbb{E} \left[ R\nabla F(\theta^t) - \Delta^t \right] \right\rangle}_{A_1} + \frac{L}{2} \eta^2 \underbrace{\mathbb{E} \left[ \left\| \Delta^t \right\|^2 \right]}_{A_2} \\
&\leq F(\theta^t) - R\eta \left( \frac{1}{2} - 30L^2 R^2 \eta^2 \right) \left\| \nabla F(\theta^t) \right\|^2 + \left( 5L^2 R^2 \eta^3 + \frac{LR}{2} \eta^2 \right) \sigma^2 \\
&\quad + \left( 30L^2 R^3 \eta^3 + R\eta \right) \sum_{k=1}^{N} \omega_k^t \sigma_k^2 + \left( \frac{L}{2} \eta^2 - \frac{\eta}{2R} \right) \sum_{k=1}^{N} \omega_k^t \mathbb{E} \left\| \sum_{r=1}^{R} \nabla F_k(\theta_k^{t,r}) \right\|^2 .
\end{aligned}
\tag{63}
$$

If $\eta < \frac{1}{8LR}$, it must be that $\frac{1}{2} - 30L^2R^2\eta^2 > 0$ and $\frac{L}{2}\eta^2 - \frac{\eta}{2R} < 0$, leading to

$$
\begin{aligned}
\mathbb{E}\left[F(\theta^{t+1})\right] \leq{} & F(\theta^t) - R\eta\left(\frac{1}{2} - 30L^2R^2\eta^2\right)\left\|\nabla F(\theta^t)\right\|^2 \\
& + \left(5L^2R^2\eta^3 + \frac{LR}{2}\eta^2\right)\sigma^2 + \left(30L^2R^3\eta^3 + R\eta\right)\sum_{k=1}^{N}\omega_k^t\sigma_k^2.
\end{aligned}
\tag{64}
$$

By rearranging and summing from $t = 0$ to $t = \mathcal{T} - 1$ we obtain

$$
\begin{aligned}
\mathbb{E}\left[F(\theta^{\mathcal{T}})\right] - F(\theta^0) \leq{} & -R\eta\left(\frac{1}{2} - 30L^2R^2\eta^2\right)\sum_{t=0}^{\mathcal{T}-1}\left\|\nabla F(\theta^t)\right\|^2 \\
& + \left(5L^2R^2\eta^3 + \frac{LR}{2}\eta^2\right)\mathcal{T}\sigma^2 + \left(30L^2R^3\eta^3 + R\eta\right)\sum_{t=0}^{\mathcal{T}-1}\sum_{k=1}^{N}\omega_k^t\sigma_k^2 \\
\leq{} & -R\eta\left(\frac{1}{2} - 30L^2R^2\eta^2\right)\mathcal{T}\min_{t\in[\mathcal{T}]}\left\|\nabla F(\theta^t)\right\|^2 \\
& + \left(5L^2R^2\eta^3 + \frac{LR}{2}\eta^2\right)\mathcal{T}\sigma^2 + \left(30L^2R^3\eta^3 + R\eta\right)\sum_{t=0}^{\mathcal{T}-1}\sum_{k=1}^{N}\omega_k^t\sigma_k^2.
\end{aligned}
\tag{65}
$$

Let $\theta^*$ denote the optimal model's parameters, i.e., $F(\theta^*) \leq F(\theta^t)\forall t \in [\mathcal{T}]$. Then

$$
\min_{t\in[\mathcal{T}]}\left\|\nabla F(\theta^t)\right\|^2 \leq \frac{1}{\mathcal{T}}\left(\frac{F(\theta^0) - F(\theta^*)}{\mathcal{A}_1} + \mathcal{A}_2\sum_{t=0}^{\mathcal{T}-1}\sum_{k=1}^{N}\omega_k^t\sigma_k^2\right) + \mathbf{\Phi},
\tag{66}
$$

where $\mathcal{A}_1 = R\eta\left(\frac{1}{2} - 30L^2R^2\eta^2\right)$, $\mathcal{A}_2 = \frac{60L^2R^3\eta^3 + 2R\eta}{R\eta(1 - 60L^2R^2\eta^2)}$ and $\mathbf{\Phi} = \frac{(10L^2R\eta^2 + L\eta)\sigma^2}{1 - 60L^2R^2\eta^2}$.

$\blacksquare$

## A.8 REGULARIZATION TERMS IN THE OBJECTIVE FUNCTION

The proposed method for estimating clients' data heterogeneity relies on the properties of gradient computed for the cross-entropy loss objective. However, the method also applies to the FL algorithms other than FedAvg, in particular those that add a regularization term to combat overfitting, including FedProx (Li et al., 2020b), FedDyn(Acar et al., 2021) and Moon (Li et al., 2021a). In the following discussion, we demonstrate that HiCS-FL remains capable of distinguishing between clients with imbalanced and balanced data when using these other FL algorithms.

### A.8.1 FEDPROX

The objective function used by FedProx (Li et al., 2020b) is

$$
\mathcal{L}_{\text{prox}}^r = \mathcal{L}_{\mathbf{CE}}^r + \frac{\mu}{2}\left\|\theta_k^{t,r} - \theta^t\right\|^2,
\tag{67}
$$

where $\theta_k^{t,r}$ is the vector of client $k$'s local model parameters in the $r$-th local epoch at global round $t$. Therefore, contribution of sample $(\mathbf{x}^{(j,n)}, y^{(j,n)})$ to the gradient of $\mathcal{L}_{\text{prox}}$ in local epoch $r$ is

$$
\frac{\partial\mathcal{L}_{\text{prox}}^{(j,n,r)}}{\partial b_i} = \frac{\partial\mathcal{L}_{\mathbf{CE}}^{(j,n,r)}}{\partial b_i} + \mu\left(b_i^{t,r} - b_i^t\right),
\tag{68}
$$

where $\mathbf{b}^{t,r} = [b_1^{t,r}, \ldots, b_C^{t,r}]$ denotes parameters of bias in the output layer of the local model, and $\mathbf{b}^t = [b_1^t, \ldots, b_C^t]$ denotes parameters of the global model at round $t$. We assume the model is trained by SGD as the optimizer, and hence

$$
b_i^{t,r} - b_i^t = b_i^{t,r-1} - \eta_t\frac{\partial\mathcal{L}_{\text{prox}}^{(j,n,r-1)}}{\partial b_i} - b_i^t = -\eta_t\frac{\partial\mathcal{L}_{\mathbf{CE}}^{(j,n,r-1)}}{\partial b_i} + (1 - \eta_t\mu)(b_i^{t,r-1} - b_i^t).
\tag{69}
$$

Therefore,

$$
\begin{aligned}
b_i^{t,r} - b_i^t &= -\eta_t \sum_{s=1}^{r-1}(1-\eta_t\mu)^{r-1-s}\frac{\partial\mathcal{L}_{\mathbf{CE}}^{(j,n,s)}}{\partial b_i} + (1-\eta_t\mu)^{r-1}(b_i^t - b_i^t) \\
&= -\eta_t \sum_{s=1}^{r-1}(1-\eta_t\mu)^{r-1-s}\frac{\partial\mathcal{L}_{\mathbf{CE}}^{(j,n,s)}}{\partial b_i},
\end{aligned}
\tag{70}
$$

and thus

$$
\frac{\partial\mathcal{L}_{\text{prox}}^{(j,n,r)}}{\partial b_i} = \frac{\partial\mathcal{L}_{\mathbf{CE}}^{(j,n,r)}}{\partial b_i} - \eta_t\mu\sum_{s=1}^{r-1}(1-\eta_t\mu)^{r-1-s}\frac{\partial\mathcal{L}_{\mathbf{CE}}^{(j,n,s)}}{\partial b_i}.
\tag{71}
$$

Taking expectation of both sides yields

$$
\begin{aligned}
\frac{1}{Bl}\sum_{j=1}^{l}\sum_{n=1}^{B}\sum_{r=1}^{R}\mathbb{E}\left[\frac{\partial\mathcal{L}_{\text{prox}}^{(j,n,r)}}{\partial b_i}\right] &= \left(-\mathbb{E}[\mathbb{I}\{i = y^{(j,n)}\}]\sum_{c\neq i}\mathcal{E}_c + \mathbb{E}[\mathbb{I}(i\neq y^{(j,n)})]\mathcal{E}_i\right) \\
&\quad \cdot \sum_{r=1}^{R}\left(1 - \eta_t\mu\sum_{s=1}^{r-1}(1-\eta_t\mu)^{r-1-s}\right) \\
&= \sum_{r=1}^{R}\left(-D_i^{(k)}\sum_{c\neq i}\mathcal{E}_c + (1-D_i^{(k)})\mathcal{E}_i\right)\left(1 - \eta_t\mu\frac{1-(1-\eta_t\mu)^{r-1}}{\eta_t\mu}\right) \\
&= \sum_{r=1}^{R}c^r\left(-D_i^{(k)}\sum_{c=1}^{C}\mathcal{E}_c + \mathcal{E}_i\right),
\end{aligned}
\tag{72}
$$

where $c^r = (1-\eta_t\mu)^{r-1} > 0$ provided $\eta_t$ and $\mu$ are sufficiently small. Therefore, the expectation of the local update of bias in the output layer satisfies

$$
\mathbb{E}\left[\Delta b_i^{(k)}\right] = C\eta_t\left(D_i^{(k)}\sum_{c=1}^{C}\mathcal{E}_c - \mathcal{E}_i\right),
\tag{73}
$$

where $C = \sum_{r=1}^{R}c^r$. Eq. (73) is similar to the expression for the expectation of the local updates of bias when applying FedAvg presented in the main paper; clearly, the analysis of HiCS-FL done in the context of FedAvg extends to FedProx.

### A.8.2 FEDDYN

For FedDyn (Acar et al., 2021), the objective function in local epoch $r$ at global round $t$ is

$$
\mathcal{L}_{\text{dyn}}^{t,r} = \mathcal{L}_{\mathbf{CE}}^{t,r} - \left\langle\nabla\mathcal{L}_{\text{dyn}}^{t-1,R}, \theta_k^{t,r}\right\rangle + \frac{\mu}{2}\left\|\theta_k^{t,r} - \theta^t\right\|^2,
\tag{74}
$$

where $R$ denotes the total number of local epochs. The first order condition for local optima implies

$$
\nabla\mathcal{L}_{\text{dyn}}^{t,r} - \nabla\mathcal{L}_{\text{dyn}}^{t-1,R} + \mu(\theta_k^{t,r} - \theta^t) = 0,
\tag{75}
$$

and, therefore,

$$
\begin{aligned}
\frac{\partial \mathcal{L}_{\text{dyn}}^{t,r}}{\partial b_i} &= \frac{\partial \mathcal{L}_{\text{dyn}}^{t-1,R}}{\partial b_i} - \mu \left( b_i^{t,r} - b_i^t \right) \\
&= \frac{\partial \mathcal{L}_{\text{dyn}}^{t-2,R}}{\partial b_i} - \mu \left( b_i^{t-1,R} - b_i^{t-1} \right) - \mu \left( b_i^{t,r} - b_i^t \right) \\
&= -\mu \sum_{\tau=1}^{t-1} \left( b_i^{\tau,R} - b_i^\tau \right) - \mu \left( b_i^{t,r} - b_i^t \right) \\
&= -\mu \sum_{\tau=1}^{t-1} \Delta b_i^\tau - \mu \left( b_i^{t,r} - b_i^t \right) \\
&= -\mu \sum_{\tau=1}^{t-1} \Delta b_i^\tau - \mu \left( -\eta_t \frac{\partial \mathcal{L}_{\text{dyn}}^{t,r-1}}{\partial b_i} + b_i^{t,r-1} - b_i^t \right) \\
&= -\mu \sum_{\tau=1}^{t-1} \Delta b_i^\tau + \mu \eta_t \left( \sum_{s=1}^{r-1} \frac{\partial \mathcal{L}_{\text{dyn}}^{t,s}}{\partial b_i} \right),
\end{aligned}
\tag{76}
$$

where $\mathbf{b}^{t,r} = [b_1^{t,r}, \ldots, b_C^{t,r}]$ denotes the bias parameters in the output layer of the local model at local epoch $r$, and where $\Delta \mathbf{b}^\tau = [\Delta b_1^\tau, \ldots, \Delta b_C^\tau]$ is the local update of the bias at round $\tau$. Since

$$
\frac{\partial \mathcal{L}_{\text{dyn}}^{t,1}}{\partial b_i} = -\mu \sum_{\tau=1}^{t-1} \Delta b_i^\tau,
\tag{77}
$$

it holds that

$$
\frac{\partial \mathcal{L}_{\text{dyn}}^{t,2}}{\partial b_i} = -\mu \sum_{\tau=1}^{t-1} \Delta b_i^\tau + \mu \eta_t \left( -\mu \sum_{\tau=1}^{t-1} \Delta b_i^\tau \right) = -\mu(1 + \mu \eta_t) \sum_{\tau=1}^{t-1} \Delta b_i^\tau
\tag{78}
$$

and

$$
\frac{\partial \mathcal{L}_{\text{dyn}}^{t,3}}{\partial b_i} = -\mu \sum_{\tau=1}^{t-1} \Delta b_i^\tau + \mu \eta_t \left( -\mu \sum_{\tau=1}^{t-1} \Delta b_i^\tau - (\mu + \mu^2 \eta_t) \sum_{\tau=1}^{t-1} \Delta b_i^\tau \right) = -\mu(1 + \mu \eta_t)^2 \sum_{\tau=1}^{t-1} \Delta b_i^\tau.
\tag{79}
$$

By induction,

$$
\frac{\partial \mathcal{L}_{\text{dyn}}^{t,r}}{\partial b_i} = -\mu(1 + \mu \eta_t)^{r-1} \sum_{\tau=1}^{t-1} \Delta b_i^\tau.
\tag{80}
$$

Therefore, the expectation of the local update of bias in the output layer at round $t$ can be computed as

$$
\mathbb{E} \left[ \Delta b_i^{(k),t} \right] = \sum_{r=1}^{R} (1 + \mu \eta_t)^{r-1} \mu \eta_t \sum_{\tau=1}^{t-1} \mathbb{E} \left[ \Delta b_i^{(k),\tau} \right]
\tag{81}
$$

$$
= \left( (1 + \mu \eta_t)^R - 1 \right) \sum_{\tau=1}^{t-1} \mathbb{E} \left[ \Delta b_i^{(k),\tau} \right].
\tag{82}
$$

Since the objective function of $\mathbb{E} \left[ \Delta b_i^{(k),1} \right]$ coincides with that of FedAvg,

$$
\mathbb{E} \left[ \Delta b_i^{(k),1} \right] = \eta_1 R \left( D_i^{(k)} \sum_{c=1}^{C} \mathcal{E}_c - \mathcal{E}_i \right),
\tag{83}
$$

where $\eta_1$ is the learning rate at global round $t = 1$. Then,

$$
\mathbb{E} \left[ \Delta b_i^{(k),2} \right] = \eta_1 R \left( (1 + \mu \eta_2)^R - 1 \right) \left( D_i^{(k)} \sum_{c=1}^{C} \mathcal{E}_c - \mathcal{E}_i \right)
\tag{84}
$$

$$
= a_1 a_2 \left( D_i^{(k)} \sum_{c=1}^{C} \mathcal{E}_c - \mathcal{E}_i \right),
\tag{85}
$$

where $a_1 = \eta_1 R$ and $a_2 = (1 + \mu\eta_2)^R - 1$. Furthermore,

$$\mathbb{E}\left[\Delta b_i^{(k),3}\right] = a_1 a_3 (1 + a_2) \left(D_i^{(k)} \sum_{c=1}^{C} \mathcal{E}_c - \mathcal{E}_i\right), \tag{86}$$

$$\mathbb{E}\left[\Delta b_i^{(k),4}\right] = a_1 a_4 (1 + a_2 + a_3 + a_2 a_3) \left(D_i^{(k)} \sum_{c=1}^{C} \mathcal{E}_c - \mathcal{E}_i\right), \tag{87}$$

and

$$\mathbb{E}\left[\Delta b_i^{(k),5}\right] = a_1 a_5 (1 + a_2 + a_3 + a_4 + a_2 a_3 + a_3 a_4 + a_2 a_3 a_4) \left(D_i^{(k)} \sum_{c=1}^{C} \mathcal{E}_c - \mathcal{E}_i\right). \tag{88}$$

By induction,

$$\mathbb{E}\left[\Delta b_i^{(k),t}\right] = \left(D_i^{(k)} \sum_{c=1}^{C} \mathcal{E}_c - \mathcal{E}_i\right) a_1 a_t \cdot \left(1 + \sum_{i=0}^{t-3} \sum_{\tau=2}^{t-1} \mathbb{I}(\tau + i < t) \prod_{i=\tau}^{\tau+i} a_s\right) \tag{89}$$

$$= a\left(D_i^{(k)} \sum_{c=1}^{C} \mathcal{E}_c - \mathcal{E}_i\right), \tag{90}$$

where $a_t = (1 + \mu\eta_t)^R - 1$ and $a = a_1 a_t \left(1 + \sum_{i=0}^{t-3} \sum_{\tau=2}^{t-1} \mathbb{I}(\tau + i < t) \prod_{i=\tau}^{\tau+i} a_s\right) > 0$. After comparing Eq. (89) with its counterpart in the case of FedAvg, we conclude that the previously presented analysis of HiCS-FL extends to FedDyn.

### A.8.3 MODEL-CONTRASTIVE FEDERATED LEARNING (MOON)

Moon (Li et al., 2021a) relies on the objective function with a contrastive term

$$\mathcal{L}_{\text{moon}} = \frac{1}{Bl} \sum_{j=1}^{l} \sum_{n=1}^{B} \mathcal{L}_{\textbf{CE}}^{(j,n)} - \mu \log \frac{\exp(\text{sim}(\mathbf{z}^{(j,n)}, \mathbf{z}_{\text{glob}}^{(j,n)})/T)}{\exp(\text{sim}(\mathbf{z}^{(j,n)}, \mathbf{z}_{\text{glob}}^{(j,n)})/T) + \exp(\text{sim}(\mathbf{z}^{(j,n)}, \mathbf{z}_{\text{prev}}^{(j,n)})/T)}, \tag{91}$$

where $\mathbf{z}^{(j,n)}$ denotes the output of the feature extractor of the local model $\theta_k^t$, $\mathbf{z}_{\text{glob}}^{(j,n)}$ is the output of the feature extractor of the global model $\theta^t$, and $\mathbf{z}_{\text{prev}}^{(j,n)}$ is the output of the feature extractor of the local model in the previous round $\theta_k^{t-1}$. Since the contrastive term does not depend on the parameters of bias in the output layer, it holds that

$$\frac{\partial \mathcal{L}_{\text{moon}}^{(j,n)}}{\partial b_i} = \frac{\partial \mathcal{L}_{\textbf{CE}}^{(j,n)}}{\partial b_i}. \tag{92}$$

Since the expectation of the local updates of bias in the output layer coincides with the one in case of FedAvg, previously presented analysis of HiCS-FL extends to Moon.

### A.9 OPTIMIZATION ALGORITHMS BEYOND SGD

Optimizers beyond SGD utilize different model update rules which in principle may lead to different properties of the local update of the bias in the output layer. However, for several variants of SGD, the properties of the local update of the bias remain such that our presented analysis still applies.

### A.9.1 SGD WITH MOMENTUM

In each local epoch $r$, SGD with momentum updates the model according to

$$m_k^{t,r} = \mu m_k^{t,r-1} + (1 - \mu)\nabla\mathcal{L}_{\textbf{CE}}^{t,r}, \tag{93}$$

$$g_k^{t,r} = m_k^{t,r}, \tag{94}$$

$$\theta_k^{t,r} = \theta_k^{t,r-1} - \eta_t g_k^{t,r}, \tag{95}$$

where $m_k^{t,r}$ denotes the momentum in the $r$-th local epoch, $\mu$ is the weight for the momentum, and $m_k^{t,1} = \nabla \mathcal{L}_{\mathbf{CE}}^{t,1}$. Then

$$\Delta \theta_k^t = -\eta_t \sum_{r=1}^{R} g_k^{t,r}, \tag{96}$$

where

$$m_k^{t,1} = \nabla \mathcal{L}_{\mathbf{CE}}^{t,1}, \tag{97}$$

$$m_k^{t,2} = \mu \nabla \mathcal{L}_{\mathbf{CE}}^{t,1} + (1-\mu)\nabla \mathcal{L}_{\mathbf{CE}}^{t,2}, \tag{98}$$

$$m_k^{t,3} = \mu \nabla \mathcal{L}_{\mathbf{CE}}^{t,2} + (1-\mu)\nabla \mathcal{L}_{\mathbf{CE}}^{t,3}$$

$$= \mu^2 \nabla \mathcal{L}_{\mathbf{CE}}^{t,1} + \mu(1-\mu)\nabla \mathcal{L}_{\mathbf{CE}}^{t,2} + (1-\mu)\nabla \mathcal{L}_{\mathbf{CE}}^{t,3}. \tag{99}$$

Therefore,

$$m_k^{t,r} = \mu^{r-1} \nabla \mathcal{L}_{\mathbf{CE}}^{t,1} + (1-\mu) \sum_{\tau=2}^{r} \mu^{r-\tau} \nabla \mathcal{L}_{\mathbf{CE}}^{t,\tau} \tag{100}$$

and thus we have

$$\Delta \theta_k^t = -\eta_t \left( \sum_{r=2}^{R} \left( \mu^{r-1} \nabla \mathcal{L}_{\mathbf{CE}}^{t,1} + (1-\mu) \sum_{\tau=2}^{r} \mu^{r-\tau} \nabla \mathcal{L}_{\mathbf{CE}}^{t,\tau} \right) + \nabla \mathcal{L}_{\mathbf{CE}}^{t,1} \right). \tag{101}$$

Similar to the discussion in the previous section,

$$\mathbb{E}\left[ \Delta b_i^{(k)} \right] = \eta_t \left( \sum_{r=2}^{R} \left( \mu^{r-1} + (1-\mu) \sum_{\tau=2}^{r} \mu^{r-\tau} \right) + 1 \right) \left( D_i^{(k)} \sum_{c=1}^{C} \mathcal{E}_c - \mathcal{E}_i \right) \tag{102}$$

$$= a \left( D_i^{(k)} \sum_{c=1}^{C} \mathcal{E}_c - \mathcal{E}_i \right) \tag{103}$$

where $a = \eta_t \left( \sum_{r=2}^{R} \left( \mu^{r-1} + (1-\mu) \sum_{\tau=2}^{r} \mu^{r-\tau} \right) + 1 \right) > 0$. Similar result is obtained when SGD applies Nesterov acceleration as long as the optimizers are not using second-order momentum.

### A.9.2 ADAM OPTIMIZER

Recall that the two observations regarding the gradient of $\mathcal{L}_{\mathbf{CE}}$ still hold when training the model with an adaptive optimizer such as Adam (Kingma & Ba, 2014). However, Adam updates the model differently from SGD. In particular, each entry of the gradient has an adaptive learning rate tied to its magnitude. With an SGD optimizer, the magnitude of the $i$-th entry of the local update of bias $\Delta \mathbf{b}^{(k)}$ is approximately proportional to the fraction of the samples with label $i$, $D_i^{(k)}$ (if $\mathcal{E}_i$ is small),

$$\mathbb{E}\left[ \Delta b_i^{(k)} \right] = \eta_t R \left( D_i^{(k)} \sum_{c=1}^{C} \mathcal{E}_c - \mathcal{E}_i \right). \tag{104}$$

However, this observation does not hold when using the Adam optimizer for the local update because each entry has a different learning rate $\eta_{t,i}$ and thus

$$\mathbb{E}\left[ \Delta b_i^{(k)} \right] = \eta_{t,i} R \left( D_i^{(k)} \sum_{c=1}^{C} \mathcal{E}_c - \mathcal{E}_i \right). \tag{105}$$

Although the magnitude of $\mathbb{E}\left[ \Delta b_i^{(k)} \right]$ is no longer approximately proportional to $D_i^{(k)}$, we can utilize the sign of $\mathbb{E}\left[ \Delta b_i^{(k)} \right]$, i.e.,

$$\text{if } D_i^{(k)} \gg D_j^{(k)}, \text{ then } \mathbb{P}\left( \mathbb{E}\left[ \Delta b_i^{(k)} \right] > 0 \right) \gg \mathbb{P}\left( \mathbb{E}\left[ \Delta b_j^{(k)} \right] > 0 \right). \tag{106}$$

Suppose client $k$ has highly imbalanced data, i.e., $H(\mathcal{D}^{(k)})$ is small. Then the maximal component $\max_i D_i^{(k)}$ is much larger than the other components; in fact, it is likely to have only one positive

component in the local update of bias $\Delta\mathbf{b}^{(k)}$. On the contrary, suppose client $u$ has balanced data and thus $H(\mathcal{D}^{(u)})$ is large. The maximal component $\max_i D_i^{(u)}$ is then very close to the other components, and it is likely to observe larger number of positive components in the local update of $\Delta\mathbf{b}^{(u)}$. While characterizing $\mathbb{P}(\mathbb{E}[\Delta b_i^{(k)}] > 0)$ appears challenging, we can empirically infer that client $u$ with more balanced data has a local update of bias $\Delta\mathbf{b}^{(u)}$ with more positive components. With

$$\hat{H}(\mathcal{D}^{(u)}) \triangleq H(\text{softmax}(\Delta\mathbf{b}^{(u)}, T)), \tag{107}$$

$$\hat{H}(\mathcal{D}^{(k)}) \triangleq H(\text{softmax}(\Delta\mathbf{b}^{(k)}, T)), \tag{108}$$

$\hat{H}(\mathcal{D}^{(u)})$ is more likely to be larger than $\hat{H}(\mathcal{D}^{(k)})$. The examples of estimated entropy when utilizing Adam as the optimizer are provided in Section. A.11.

## A.10 VISUALIZATION OF DATA PARTITIONS

To generate non-IID data partitions we follow the strategy in (Yurochkin et al., 2019), utilizing Dirichlet distribution with different concentration parameters $\alpha$ to control the heterogeneity levels. In particular, the number of samples with label $i$ owned by client $k$ is set to $\frac{X_i^{(k)} N_i}{\sum_{j=1}^N X_i^{(j)}}$, where $X_i^{(1)}, \ldots, X_i^{(N)}$ are drawn from $\text{Dir}(\alpha)$ and $N_i$ denotes the total number of samples with label $i$ in the overall dataset. For the setting with multiple $\alpha$, we divide the overall training set into $|\alpha|$ equal parts and generate data partitions according to the method above. Figures 7 and 8 illustrate the class distribution of local clients by displaying the number of samples with different labels; colors distinguish between magnitudes – the darker the color, the more samples are in the class.

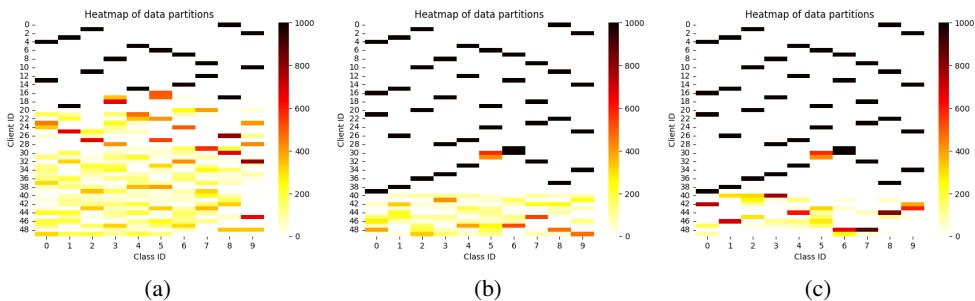

Figure 7: Results on CIFAR10. Training data is split into 50 partitions according to a Dirichlet distribution (50 clients). The concentration parameter is as follows: (1) $\alpha \in \{0.001, 0.01, 0.1, 0.5, 1.0\}$; (2) $\alpha \in \{0.001, 0.002, 0.005, 0.01, 0.5\}$; (3) $\alpha \in \{0.001, 0.002, 0.005, 0.01, 0.1\}$. The figures (a), (b) and (c) correspond to settings (1), (2) and (3), respectively.

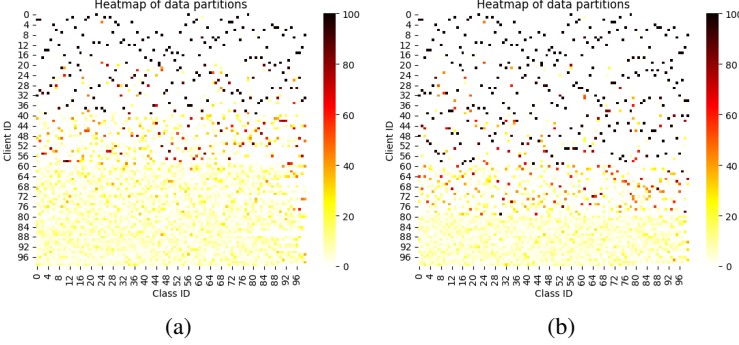

Figure 8: Results on CIFAR100. Training data is split into 100 partitions according to Dirichlet distribution (100 clients). The concentration parameter is varied as follows: (1) $\alpha \in \{0.001, 0.01, 0.1, 0.5, 1.0\}$; (2) $\alpha \in \{0.001, 0.005, 0.01, 0.1, 1.0\}$. The figures (a) and (b) correspond to settings (1) and (2), respectively.

### A.11 EXAMPLES OF ESTIMATED ENTROPY

To further illustrate the proposed framework, here we show a comparison between the estimated entropy of data label distribution and the true entropy. Specifically, Figures 9 and 10 show that the entropy estimated by the proposed method is close to the true entropy; the experiments were conducted on FMNIST and CIFAR100, using SGD and Adam as optimizers, respectively. As stated in Theorem 1, the clients with larger true entropy are likely to have lager estimated entropy. In case where the model is trained with Adam, estimated entropy of data label distribution is not as accurate as in the case of using SGD. Figures 11 and 12 compare the performance of estimating entropy with SGD and Adam optimizers for the same setting of $\alpha$. Notably, as shown in the figures, the method is capable of distinguishing clients with extremely imbalanced data from those with balanced data.

### A.12 ADDITIONAL EXPERIMENTAL RESULTS

In this section, we report average training loss and standard deviation obtained by running experiments on FMNIST, CIFAR10 and CIFAR100 datasets using the same settings as those described in Section 4 in the main paper, aiming to characterize convergence of the global model trained with different methods. Figures 13, 15 and 17 show that the average local training loss follows a similar pattern as the test accuracy of the global model, and that HiCS-FL (ours) outperforms all other methods. Figures 14 and 16 shows that HiCS-FL has the lowest standard deviation in almost all settings. Standard deviation of HiCS-FL increases during training since the sampling policy $\pi^t$ converges to random sampling as described in Section 3.4 in the main paper. However, heterogeneity of FMNIST dataset does not have a major effect on standard deviation, likely due to relative simplicity of FMNIST as compared to CIFAR10.

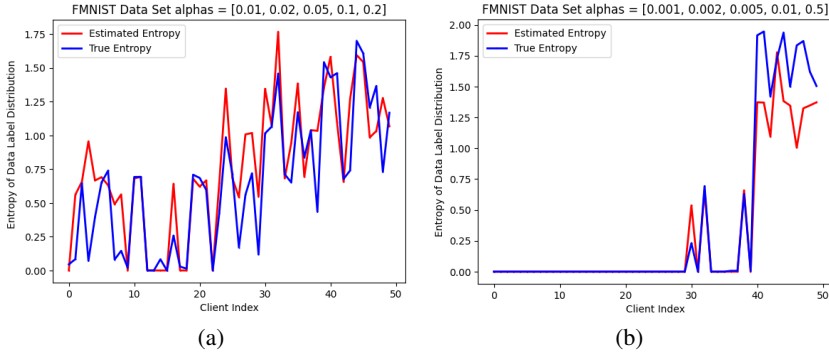

Figure 9: The estimated entropy of data label distribution in experiments on FMNIST with SGD as the optimizer. The parameter $\alpha$ for the two figures: (a) $\alpha \in \{0.01, 0.02, 0.05, 0.1, 0.2\}$; (b) $\alpha \in \{0.001, 0.002, 0.005, 0.01, 0.5\}$

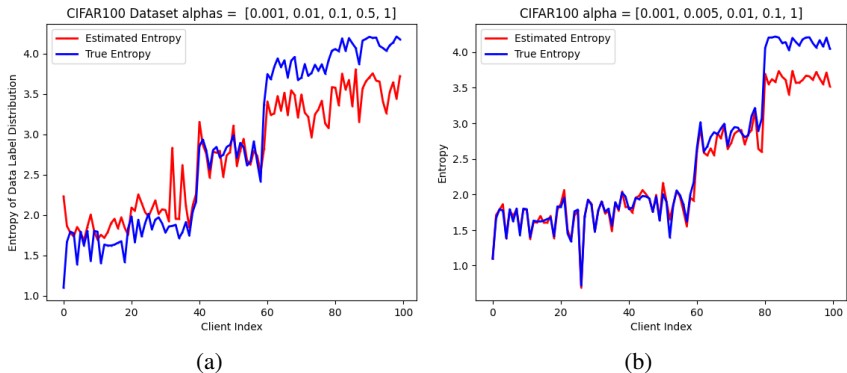

Figure 10: The estimated entropy of data label distribution in experiments on CIFAR100 with Adam as the optimizer. The parameter $\alpha$ for the two figures: (a) $\alpha \in \{0.001, 0.01, 0.1, 0.5, 1.0\}$; (b) $\alpha \in \{0.001, 0.005, 0.01, 0.1, 1.0\}$.

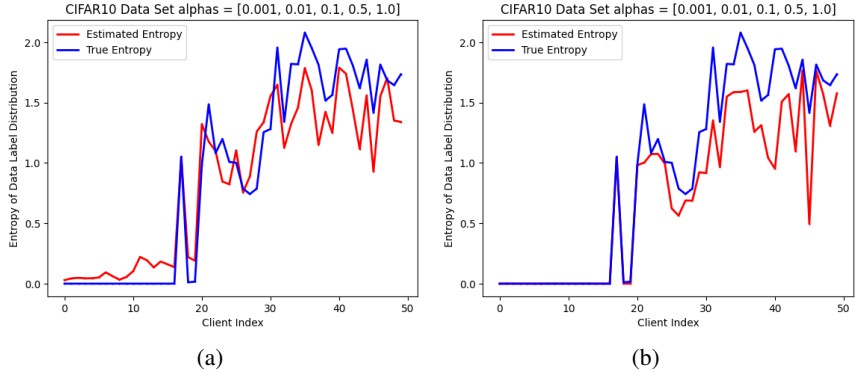

(a)             (b)

Figure 11: The estimated entropy of data label distribution in experiments on CIFAR10 with $\alpha \in \{0.001, 0.01, 0.1, 0.5, 1.0\}$. (a) The result of the experiments using SGD as the optimizer. (b) The result of the experiments using Adam as the optimizer.

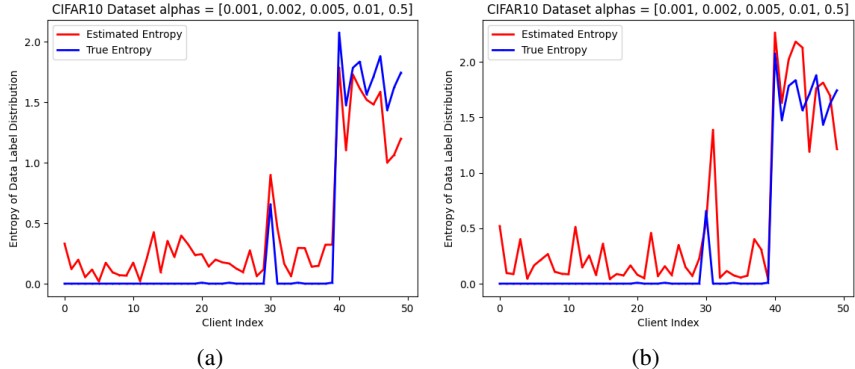

(a)             (b)

Figure 12: The estimated entropy of data label distribution in experiments on CIFAR with $\alpha \in \{0.001, 0.002, 0.005, 0.01, 0.5\}$. (a) The result of the experiments using SGD as the optimizer. (b) The result of the experiments using Adam as the optimizer.

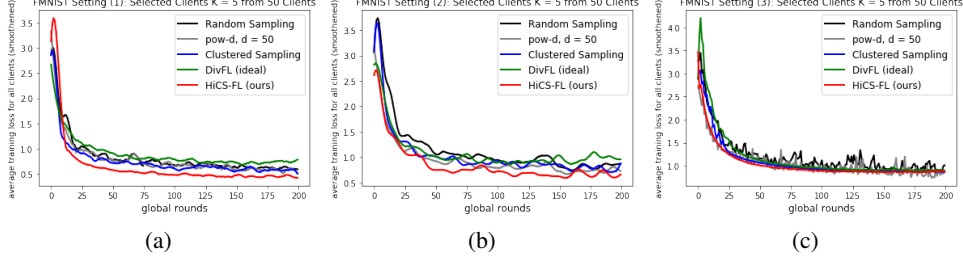

(a)           (b)           (c)

Figure 13: Average training loss of HiCS-FL and the competing methods on 3 groups of data partitions of FMNIST; the concentration parameter $\alpha$ follows the stated settings (1)-(3).

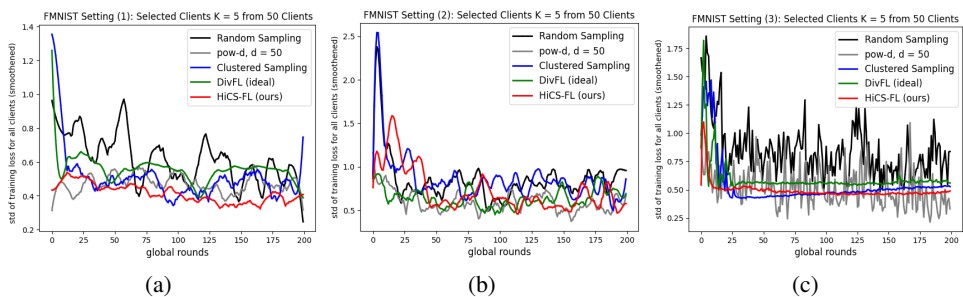

Figure 14: Standard deviation of training loss of HiCS-FL and the competing methods on 3 groups of data partitions of FMNIST; the concentration parameter $\alpha$ follows the stated settings (1)-(3).

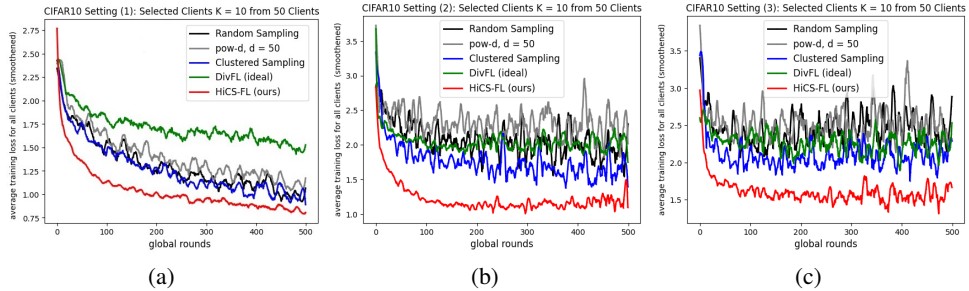

Figure 15: Average training loss of HiCS-FL and the competing methods on 3 groups of data partitions of CIFAR10; the concentration parameter $\alpha$ follows the stated settings (1)-(3).

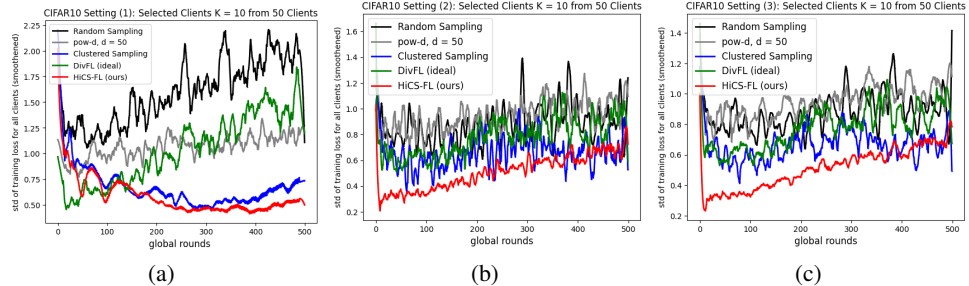

Figure 16: Standard deviation of training loss of HiCS-FL and the competing methods on 3 groups of data partitions of CIFAR10; the concentration parameter $\alpha$ follows the stated settings (1)-(3).

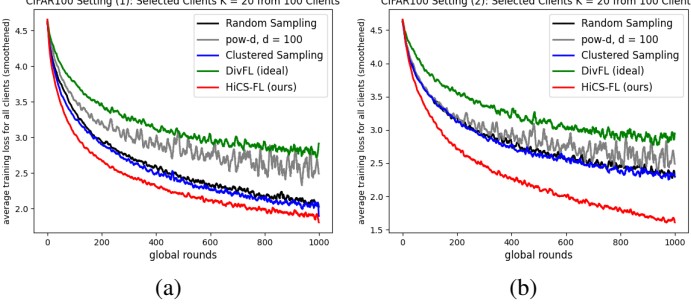

Figure 17: Average training loss of HiCS-FL (in red) and the competing methods on 2 groups of data partitions of CIFAR100; the concentration parameter $\alpha$ follows the stated settings (1)-(2).

### A.13 COMPUTATIONAL AND COMMUNICATION COMPLEXITY

We compare the communication and computational costs of HiCS-FL with those of the competing methods, including Power of Choice (pow-d) (Cho et al., 2020), Clustered Sampling (Fraboni et al., 2021) and DivFL (Balakrishnan et al., 2022), and map them against random sampling. In its ideal setting, pow-d selects $K$ clients with the largest local validation loss among all $N$ clients. To compute the local validation loss at the beginning of a global training round $t$, the server must send the global model to all clients. Compared to the random sampling strategy where the global model is sent to only $K$ clients, pow-d must transmit additional $(N - K)|\theta^t|$ model parameters. Moreover, pow-d requires all clients to compute validation loss of the global model $\theta^t$ on local datasets, which incurs additional $\mathcal{O}(N|\theta^t|)$ computations. While communication requirements of Clustered Sampling do not exceed those of random sampling, the server must run a clustering algorithm on the local updates of dimension $|\theta^t|$ (the same as gradients). DivFL relies on maximizing a submodular function to select the most diverse clients based on all clients' gradients, leading to a transmission overhead and additional computation involving $|\theta^t|$-dimensional gradients. In our experiments, DivFL has consistently required the longest training time and memory usage due to its dependence on the submodularity maximizer. Our proposed method, HiCS-FL, does not require any additional transmission of model parameters; furthermore, in HiCS-FL the server clusters clients based on their local updates of the bias in the output layer, which is low-dimensional and model-agnostic. Overall, HiCS-FL requires negligible computational overhead to significantly improve the performance of non-iid Federated Learning.

