# OpenReview forum: "Accelerating Non-IID Federated Learning via Heterogeneity-Guided Client Sampling"
_ICLR.cc/2024/Conference — Submitted to ICLR 2024_

### Official Review · Reviewer_BfMQ · 2023-10-23

**Soundness:** 3 good
**Presentation:** 2 fair
**Contribution:** 3 good
**Rating:** 8
**Confidence:** 3

**Summary:**

This paper introduces a novel method for federated learning on non-iid data, with a particular focus on client sampling. The proposed method expands upon clustered sampling, making it hierarchical. Existing methods cluster clients into groups based on the similarity of their local updates, selecting one client per group to avoid redundant updates. However, the proposed method differs in several ways. Firstly, the server estimates the heterogeneity of each client's data based on the entropy of soft-max outputs. This estimation characterizes each client group based on average heterogeneity. In the proposed hierarchical scheme, client groups are initially sampled based on their average heterogeneity, with groups demonstrating higher heterogeneity having a greater chance of selection. Following this, clients are sampled from the selected groups based on client weights p_k (batch size). The paper also provides a convergence analysis. Experimental results on three image datasets (FMNIST, CIFAR-10, and CIFAR-100) validate the effectiveness of the proposed method.

**Strengths:**

**Originality**: The proposed client sampling scheme, which selects client groups based on data heterogeneity and then selects clients based on client weights, is new.

**Quality**: The quality of the proposed work is satisfactory overall, presenting some novel ideas that are quantitatively evaluated on multiple datasets. These ideas have proven to be more effective compared to some other state-of-the-art methods.

**Clarity**: The work's motivation is clear. Sophisticated client sampling is indeed necessary for FL with a large number of clients, where standard random sampling may not always be the optimal solution. The relationship to prior work, especially the clustered sampling strategy (Fraboni et al., 2021), is clearly stated.

**Significance**: The proposed method significantly improves training speed, ranging from 1.63x to 7.3x across multiple datasets.

**Weaknesses:**

**Clarity**: The organization of Section 3 could be improved. It currently includes background ideas, existing approaches, and the proposed method, making it difficult to identify the novel aspects of this work. A possible solution could be to split it into two sections: "Preliminaries" and "Proposed Method".

**Applications to other tasks**: The proposed method has only been proven effective on image classification tasks (FMNIST, CIFAR-10, and CIFAR-100). It remains unclear whether the method can be extended to non-classification tasks (e.g., regression) or non-image tasks (e.g., NLP).

**Comparisons with other state-of-the-art methods without client sampling**: The significance of the proposed method over existing client sampling-based approaches such as pow-d, CS, and DivFL is clear. However, the necessity for client sampling is not. There are numerous methods addressing non-iidness in FL, not just FedProx, but also Scaffold [Karimireddy et al. 2020], Scaffnew [Mishchenko et al., 2022], FedNova [Wang et al., 2020], FedOpt [Reddi et al., 2021]. The contribution of the proposed method would be clearer if it were compared against these non-sampling-based methods quantitatively in terms of training speeds and communication/computation overheads.

**Questions:**

Based on the weaknesses, I propose the following questions:

- Can the proposed method be evaluated on tasks other than image classification? For example, the Shakespeare (next work prediction) in the LEAF dataset (https://leaf.cmu.edu/) is a task commonly used in FL.
- Can the proposed method be compared with other non-sampling based approaches such as Scaffold, NedNova, or FedOpt that outperformed FedProx?

---

> ### Author Response · Authors · 2023-11-15
> **Response to Reviewer BfMQ22**
>
> We appreciate the reviewer’s positive feedback regarding the paper’s novelty, and the proposed method’s effectiveness and significance. Below we aim to address the reviewer’s concerns and questions.
>
> **3.1Q**: *“The organization of Section 3 could be improved. It currently includes background ideas, existing approaches, and the proposed method, making it difficult to identify the novel aspects of this work.”*
>
> **3.1A**: Thanks for the suggestions! We are committed to following up and improving the paper’s presentation and organization. We have already taken steps to clarify the novel aspects of the work – please see our response to reviewer V18X30 (parts of which we incorporated in the revised paper).
>
> **3.2Q**: *“Can the proposed method be evaluated on tasks other than image classification”.*
>
> **3.2A**: As stated in Section 3.2.1, our method is applicable to any multi-class classification tasks. Moreover, the method is model-agnostic since we make no assumptions on the model architecture. While our evaluation has focused on the widely used image datasets (FMNIST and CIFAR-10/100), the method could in principle be applied to e.g. transformer-based models used for text classification.
>
> **3.3Q**: *“Can the proposed method be compared with other non-sampling based approaches such as Scaffold, NedNova, or FedOpt that outperformed FedProx?”*
>
> **3.3A**: Scaffold, FedNova and FedOpt address the data heterogeneity problem in FL using regularization or adaptive optimizers, which is indeed a different strategy than client selection utilized by the methods [1]-[4]. We view the two sets of techniques as being complementary rather than competing with each other. It would certainly be very interesting to further explore integration of these two sets of approaches but a comprehensive study on that topic would be rather involved and goes beyond the scope of the current paper.
>
> **3.4Q**: *“The contribution of the proposed method would be clearer if it were compared against these non-sampling-based methods quantitatively in terms of training speeds and communication/computation overheads”*
>
> **3.4A**: As stated in the paper (please see the second paragraph in Section 4.2 and Table 1), the communication cost of our method is identical to the communication cost of FedAvg – our method does not require any additional information and thus incurs no communication overhead. Regarding the computation overhead: The only additional computation is to estimate data heterogeneity and perform clustering based on the bias updates. These operations are negligible compared to the computational efforts required to update the model. It is helpful that the reviewer raised this question, we will make sure to highlight these points in the revised paper; currently, most of the discussion on this topic is in Appendix A.13.
>
>
> References:
>
> [1] Chen W, Horvath S, Richtarik P. Optimal client sampling for federated learning[J]. arXiv preprint arXiv:2010.13723, 2020.
>
> [2] Cho Y J, Wang J, Joshi G. Client selection in federated learning: Convergence analysis and power-of-choice selection strategies[J]. arXiv preprint arXiv:2010.01243, 2020.
>
> [3] Fraboni Y, Vidal R, Kameni L, et al. Clustered sampling: Low-variance and improved representativity for clients selection in federated learning[C]//International Conference on Machine Learning. PMLR, 2021
>
> [4] Balakrishnan R, Li T, Zhou T, et al. Diverse client selection for federated learning via submodular maximization[C]//International Conference on Learning Representations. 2022.

---

> > ### Comment · Reviewer_BfMQ · 2023-11-15
> > **Thank you**
> >
> > Thank you for the response! It solved some of my initial concerns. I understand that the proposed method can in principle be used for text/word classification tasks, but if the proposed method is shown to perform well on such NLP tasks empirically, I think the significance of the work could have been evaluated higher.

---

> > > ### Author Response · Authors · 2023-11-17
> > > **Additional Experiments in Text Classification**
> > >
> > > Following up on the reviewer’s comment that the experimental results on NLP tasks would further strengthen the significance of our work, we conducted additional experiments involving text classification on the THUC news dataset (https://github.com/thunlp/THUCTC) in Chinese language with 10 labels. There are 18000 samples in the training set, and 10000 samples in each the validation and test set.  For consistency with the main paper, we allocated data to 50 clients by emulating heterogeneous data distributions scenarios with parameter of $\alpha$ set to: (1) {0.001, 0.01, 0.1, 0.2,1}; (2) {0.001, 0.002, 0.01, 0.1,0.5}; and (3) {0.001, 0.002, 0.005, 0.01, 0.1}. We trained **TextRNNs** [1] with BiLSTM architecture (very different from the CNNs used in our prior experiments) as the classifiers with Adam optimizers. Further details of the experimental settings could be provided in the appendix of the revised submission. The results, reported in Table 3 below, show that our method outperforms baselines in all the settings, demonstrating efficacy of our proposed algorithm in an NLP task. For completeness, we also provide Table 4 showing the number of global rounds needed for these experiments to achieve 80% test accuracy.
> > >
> > > Table 3: Test accuracy of TextRNN trained for a text classification task on THUC news dataset after 100 rounds of training.
> > >
> > > |  | setting (1) | setting (2) | setting (3) |
> > > |---|:---:|:---:|:---:|
> > > | Random | 78.9 | 74.9 | 72.7 |
> > > | pow-d | 80.0 | 75.4 | 66.5 |
> > > | CS | 80.6 | 82.8 | 79.4 |
> > > | DivFL | 73.0 | 68.9 | 72.1 |
> > > | FedCor ($\beta = 0.9$)  | 81.2 | 81.3 | 76.4 |
> > > | HiCS-FL  | **83.2** | **83.9** | **79.7** |
> > >
> > > Table 4: The number of global rounds needed to achieve **80%** test accuracy. The settings are the same as in the experiments reported in Table 3.
> > >
> > > |  | setting (1) | setting (2) | setting (3) |
> > > |---|:---:|:---:|:---:|
> > > | Random | 56 | 83 | 179 |
> > > | pow-d | 67 | 109 | 159 |
> > > | CS | 48 | 74 | 107 |
> > > | DivFL | 189 | 289 | 355 |
> > > | FedCor  | 64 | 100 | 131 |
> > > | HiCS-FL  | **25** | **27** | **91** |
> > >
> > >
> > > Reference:
> > >
> > > [1] Liu P, Qiu X, Huang X. Recurrent neural network for text classification with multi-task learning[J]. arXiv preprint arXiv:1605.05101, 2016.

---

> > > > ### Comment · Reviewer_BfMQ · 2023-11-20
> > > > **Thank you**
> > > >
> > > > Thank you for the additional experiment! It looks promising, and based on that I would like to upgrade my rating.

---

### Official Review · Reviewer_V18X · 2023-10-30

**Soundness:** 2 fair
**Presentation:** 2 fair
**Contribution:** 2 fair
**Rating:** 5
**Confidence:** 3

**Summary:**

This paper introduces a novel client selection approach known as HiCS-FL (Federated Learning via Hierarchical Clustering Sampling). HiCS-FL utilizes client updates sent to the server's network output layer to estimate the statistical heterogeneity of client data. Using this information, it conducts clustering and client sampling. The paper provides an in-depth analysis of HiCS-FL's capability to assess the heterogeneity across various datasets and elucidates the convergence behavior during the deployment of this client selection method.

**Strengths:**

1. The paper addresses an important problem in the FL setting.

2. The proposed method outperforms some existing methods in the experiment section.

3. The paper is well-written and easy to understand.

**Weaknesses:**

1. The convergence analysis provided lacks depth and originality. It appears to closely resemble the analysis conducted in prior research on the conventional federated learning (FL) setting. Consequently, it does not offer any unique insights specific to the proposed method.

2. The proposed method needs to address privacy concerns.

3. The comparison methods employed in the study are both limited in scope and outdated. It is strongly recommended that the authors thoroughly explore recent and state-of-the-art methodologies to offer a more comprehensive and up-to-date evaluation of their proposed approach.

**Questions:**

Please see the weakness.

---

> ### Author Response · Authors · 2023-11-15
> **Response to Reviewer V18X [1/2]**
>
> We appreciate the reviewer’s positive feedback regarding the strengths of our paper. However, we respectfully disagree with the comments implying that our method lacks depth and originality, and that the baselines in the experiments are outdated. While the reviewer did not bring up specific questions which we could address point-by-point, we will nevertheless try our best to address the reviewer’s comments in broad strokes.
>
> **2.1Q**: *“The convergence analysis provided lacks depth and originality. It appears to closely resemble the analysis conducted in prior research on the conventional federated learning (FL) setting.”*
>
> **2.1A**: We respectfully disagree, the considered problem setting and our approach bring forth distinctive challenges and solutions that, to the best of our knowledge, are not encountered in related prior work. For clarity, we need to state a brief summary of our contributions:
>
> (1) At the core of our proposed method is a fundamental property of the update of the bias in the output layer of a broad class of neural networks; our analysis of this fundamental property, presented in Section 3.2 and Appendix A.3 and A.4, is likely of independent interest.
>
> (2) Based on the aforementioned property of (local) updates, we developed a novel method for estimating the heterogeneity of clients’ data distribution. Our data heterogeneity estimation method requires no information from the clients other than their regular model updates – the information already collected by the server in a FL system. The method is theoretically analyzed in Appendix A.6.
>
> (3) We developed and theoretically analyzed a novel adaptive client sampling method, HiCS-FL, which accelerates the convergence of FedAvg in settings where the clients have data with different levels of heterogeneity. The convergence analysis required a non-trivial extension of the techniques used in previous studies. The closest work to our paper [4], cited in Appendix A.7,  provides a convergence analysis of an unbiased sampling strategy proposed there (please see the definition in Section 2 on the related work). Our work provides convergence analysis for a much more general arbitrary sampling strategy; for details, and for insight into the acceleration of the proposed HiCS-FL, please see Theorem 2.
>
> (4) We conducted a comprehensive analysis of the proposed client sampling method used within several frameworks (specifically, FedAvg, FedProx, FedDyn and Moon) and trained with several different optimizers (specifically, SGD, SGD-momentum and Adam). The relevant derivations can be found in Appendix A.8 and A.9.
>
> The contributions (1) - (4) are distinct from the existing work on client sampling in FL. We would appreciate it if the reviewer points out specific references so that we could in more concrete terms compare and contrast our work to the prior work the reviewer is referring to.
>
>
> **2.2Q**: “The proposed method needs to address privacy concerns.”
>
> **2.2A**: Our work is driven by the need for client sampling strategies suitable for settings where the clients’ data is heterogeneous. In developing such sampling strategies, we do not require any information beyond the information already collected by the servers in federated learning systems. Therefore, our method does not introduce any additional privacy risks.

---

> ### Author Response · Authors · 2023-11-15
> **Response to Reviewer V18X [2/2]**
>
> **2.3Q**: *“The comparison methods employed in the study are both limited in scope and outdated. It is strongly recommended that the authors thoroughly explore recent and state-of-the-art methodologies to offer a more comprehensive and up-to-date evaluation of their proposed approach.”*
>
> **2.3A**: The baselines in our experiments include pow-d (2020 AISTAT) [5], clustered sampling (2021 ICML) [6], and DivFL (2022 ICLR) [7], state-of-the-art methods for client selection in FL systems. Moreover, these three methods are explainable and come with theoretical analysis/guarantees, as the method in our paper does. Following the suggestion of reviewer yWNZ, we added one more baseline to the experiments, FedCor (2022 CVPR) – please see our response to reviewer yWNZ. There indeed exist a number of other client selection methods, e.g., those that rely on reinforcement learning techniques. However, those methods tend to be empirical in nature, and challenging to analyze/interpret. That being said, we would certainly welcome suggestions from the reviewer regarding additional baselines, especially if said baselines are analytically well-grounded.
>
> Reference:
>
> [1] Li T, Sahu A K, Zaheer M, et al. Federated optimization in heterogeneous networks[J]. Proceedings of Machine learning and systems, 2020
>
> [2] Acar D A E, Zhao Y, Navarro R M, et al. Federated learning based on dynamic regularization[J]. arXiv preprint arXiv:2111.04263, 2021.
>
> [3] Li Q, He B, Song D. Model-contrastive federated learning[C]//Proceedings of the IEEE/CVF conference on computer vision and pattern recognition. 2021
>
> [4] Chen W, Horvath S, Richtarik P. Optimal client sampling for federated learning[J]. arXiv preprint arXiv:2010.13723, 2020.
>
> [5] Cho Y J, Wang J, Joshi G. Client selection in federated learning: Convergence analysis and power-of-choice selection strategies[J]. arXiv preprint arXiv:2010.01243, 2020.
>
> [6] Fraboni Y, Vidal R, Kameni L, et al. Clustered sampling: Low-variance and improved representativity for clients selection in federated learning[C]//International Conference on Machine Learning. PMLR, 2021
>
> [7] Balakrishnan R, Li T, Zhou T, et al. Diverse client selection for federated learning via submodular maximization[C]//International Conference on Learning Representations. 2022.

---

> ### Author Response · Authors · 2023-11-20
> **Kindly Reminder**
>
> As the rebuttal period draws to a close, we would really appreciate receiving your valuable input.
> We are encouraged by the positive and constructive dialogue with reviewer BfMQ (please see our
> posted exchange). Any suggestions that you may have would greatly contribute to further enhancing
> our work!

---

> > ### Comment · Reviewer_V18X · 2023-11-21
> > **Thank you for the reponse**
> >
> > I appreciate the author's response, and I have adjusted my rating accordingly. I think this paper is on the borderline of acceptance, and its ultimate acceptance may also be influenced by the quality and relevance of other submitted papers in this context.

---

### Official Review · Reviewer_yWNZ · 2023-11-02

**Soundness:** 3 good
**Presentation:** 3 good
**Contribution:** 3 good
**Rating:** 5
**Confidence:** 3

**Summary:**

This paper introduced a novel method of selecting clients in federated learning by clustering clients from their estimated data heterogeneity. Specifically, this paper found the relationship between the last layer's bias and the distribution of all labels in a given client's local data. By applying the Hierarchical Clustered Sampling technique, the server can compute this estimation for all clustered groups from a random sampling. By picking clients with the highest estimation, the server will aggregate clients with more balanced labels without knowing the exact distribution of the local data. This method achieved faster convergence and less overhead in extensive experiments compared with current works (FedProx) in this field on public datasets (FMNIST, CIFAR-10, and CIFAR-100).

**Strengths:**

++ This paper introduced a novel method to solve client selection problems

++ This paper provided a detailed convergence analysis

**Weaknesses:**

-- This paper didn't provide enough experiments to support this method. Please add experiments for more complicated networks, current test accuracy is too low to verify the efficiency of the algorithm.

-- This paper didn't evaluate cases with client inaccessibility issues, though which were mentioned in the abstract. Please provide experiments when a given fraction of clients are unavailable and show the performance of your methods with other STOA methods.

-- Writing issues: some part of this paper is a little confusing and there are some typos in certain figures. For example, in Figure 6, the y-axis label is too long, explain it in the caption of the figure.

**Questions:**

- My greatest concern is that this method may not be extended to cases other than SGD as mentioned in this paper. Even though there are proofs in the appendix, no experiments are provided to deomonstrate that. The appendix mentioned your method could be extended to the non-SGD optimizers. Please provide experiments to benchmark with STOA methods like FedCor with optimizers like Adam. The reason I recommended FedCor is that it doesn't need a loss from all clients and shows good performances in terms of convergence speed and final accuracy.

---

> ### Author Response · Authors · 2023-11-15
> **Response to Reviewer yWNZ [1/2]**
>
> We appreciate the reviewer’s positive comments regarding the novelty of our method and the detailed theoretical analysis that we provide. We would like to highlight another contribution of the manuscript: A novel method for estimating the heterogeneity of clients’ data distribution without requiring any information from the clients other than the local updates (essentially, the method requires updates that are typically already collected by the server in FL systems).
>
> The reviewer also raised several concerns which we attempt to address in details below.
>
> **1.1Q**. *“My greatest concern is that this method may not be extended to cases other than SGD as mentioned in this paper.”*
>
> **1.1A**: Actually, the experiments on CIFAR-10/100 data presented in the main paper are conducted using Adam optimizer, as stated in Section 4.1 (please see the first sentence of the second and third paragraph there). Moreover, Appendix A.9 “Optimization Algorithms Beyond SGD” presents an analysis of the proposed scheme for (a) SGD with momentum and (b) Adam optimizer. This analysis is omitted from the main paper only due to space limitations. Our apologies if our statements in the main paper were insufficiently clear, we will certainly clarify in the revised manuscript that the method applies to optimizers beyond SGD.
>
> **1.2Q**: *“This paper didn't provide enough experiments to support this method. Please add experiments for more complicated networks, current test accuracy is too low to verify the efficiency of the algorithm. Please provide experiments to benchmark with STOA methods like FedCor with optimizers like Adam. ”*
>
> **1.2A**: We thank the reviewer for the suggestions regarding incorporating among baselines other model architectures and the SOTA methods such as FedCor [2]. Following up on this request, we conducted additional experiments and are reporting the results below. However, please note that the performance of FedCor, as reported in [2], is comparable to the performance of HiCS-FL, as reported in our manuscript, even though our manuscript explores heterogeneous data settings which are considerably more challenging. Indeed, data heterogeneity, rather than the capacity of the networks, is the main reason for the relatively low accuracy (i.e., the low accuracy in data heterogeneous settings is caused by the client-drift problem, as previously argued in e.g. [1]). As illustrated in Appendix A.10, clients’ local datasets may be small; in extreme cases, there could be only one or two classes present in those datasets. In such settings, local training of deep models (e.g., ResNet) will often lead to overfitting, ultimately causing relatively low test accuracy.
>
> The new experiments: For a fair comparison of our method to FedCor we ran experiments with ResNet20 and CNNs on CIFAR-10/100, where the models are trained using the Adam optimizer; the FedCor’s code is obtained from the official repository (https://github.com/Yoruko-Tang/FedCor/tree/fedcor_new). The results are reported in Table 1 and Table 2 below. As can be seen there, using a deeper network does not improve the test accuracy in the settings with extremely heterogeneous client data (i.e., overfitting still happens). Nevertheless, since our method (HiCS-FL) selects clients with more balanced data, it succeeds to a degree to mitigate overfitting and generally outperforms the competing methods. FedCor achieves competitive performance only in setting (3), which involves CNNs.
>
> Table 1: Test accuracy of ResNet20 trained for a classification task on CIFAR10. The FL system has 50 clients, the number of global rounds is set to 500, two local epochs. As in the main papr, we consider 3 heterogeneity settings parameterized via the values of the Dirichlet distribution parameter alpha: (1) {0.001, 0.01, 0.1, 0.5, 1}; (2) {0.001, 0.002, 0.005, 0.01, 0.5]}; (3) {0.001,0.002,0.005,0.01,0.1}. Note that beta in FedCor is the annealing coefficient controlling the sampling strategy [2]; we set it to the recommended value.
>
> | | setting (1) | setting (2) | setting (3) |
> |---|:---:|:---:|:---:|
> | Random | 48.7 | 33.3 | 22.5 |
> | pow-d | 41.9 | 40.9 | 30.3 |
> | CS | 60.2 | 31.8 | 28.5 |
> | DivFL | 39.6 | 26.3 | 22.3 |
> | FedCor ($\beta = 0.9$) | 53.7 | 43.1 | 25.5 |
> | HiCS-FL  | **61.7** | **50.2** | **35.3** |
>
>
> Table 2: Test accuracy of CNNs trained for a classification task on CIFAR10. The other settings are the same as in Table 1.
>
> | | setting (1) | setting (2) | setting (3) |
> |---|:---:|:---:|:---:|
> | Random | 55.6 | 36.1 | 35.1 |
> | pow-d | 52.4 | 34.8 | 33.6 |
> | CS | 61.4 | 46.3 | 43.9 |
> | DivFL | 43.6 | 35.6 | 27.1 |
> | FedCor ($\beta = 0.9$)  | **64.3** | 52.6 | 42.1 |
> | HiCS-FL  | 63.9 | **61.7** | **50.5** |

---

> ### Author Response · Authors · 2023-11-15
> **Response to Reviewer yWNZ [2/2]**
>
> In addition to the accuracy of the trained models, reported in the main paper, in the appendix we provide the training loss and its variance. These results demonstrate that our HiCS-FL exhibits smoother training with lower variance than the competing methods. We also conducted a number of experiments showing the effects of data distribution heterogeneity on the performance of our method (please see Appendix A.12). Finally, we ran extensive experiments to verify Assumption 1 in Section 3.1 (please Appendix A.2). In all, we believe the paper offers comprehensive experiments that demonstrate the performance and provide insight into the proposed framework.
>
> **1.3Q**: *“This paper didn't evaluate cases with client inaccessibility issues, though which were mentioned in the abstract. Please provide experiments when a given fraction of clients are unavailable and show the performance of your methods with other STOA methods”*
>
> **1.3A**: The reviewer is likely referring to the following sentence in Abstract: “Particularly challenging are the settings where due to resource constraints only a small fraction of clients can participate in any given round of FL.” What we mean is that in order to satisfy communication constraints, the server selects only a subset of clients out of the entire cohort of clients – more precisely, the assumption is that the number of available clients exceeds the limit imposed by bandwidth constraints, and that the server selects a subset of the available clients. This point is further clarified in Introduction (the 1st sentence of the 3rd paragraph). Similar settings are considered by the authors of the baselines [2-5] to which we compare our proposed method. While the scenario where, unbeknownst to the server, a selected client is unavailable has not been within the scope of our work thus far, the reviewer makes a valid suggestion – it would indeed be interesting to consider it as part of future work.
>
> **1.4Q**: *“some parts of this paper are a little confusing and there are some typos in certain figures. For example, in Figure 6, the y-axis label is too long, explain it in the caption of the figure”*
>
> **1.4A**: Good catch, thanks for pointing this out! We are committed to improving the readability and fixing such errors in the final version of the manuscript, should it be accepted.
>
> Reference:
>
> [1] Karimireddy S P, Kale S, Mohri M, et al. Scaffold: Stochastic controlled averaging for federated learning[C]//International conference on machine learning. PMLR, 2020: 5132-5143.
>
> [2] Tang M, Ning X, Wang Y, et al. FedCor: Correlation-based active client selection strategy for heterogeneous federated learning[C]//Proceedings of the IEEE/CVF Conference on Computer Vision and Pattern Recognition. 2022
>
> [3] Cho Y J, Wang J, Joshi G. Client selection in federated learning: Convergence analysis and power-of-choice selection strategies[J]
>
> [4] Fraboni Y, Vidal R, Kameni L, et al. Clustered sampling: Low-variance and improved representativity for clients selection in federated learning[C]//International Conference on Machine Learning. PMLR, 2021: 3407-3416.
>
> [5] Balakrishnan R, Li T, Zhou T, et al. Diverse client selection for federated learning via submodular maximization[C]//International Conference on Learning Representations. 2022

---

> > ### Author Response · Authors · 2023-11-18
> > **Additional Experiments in Text Classification**
> >
> > Following up on the reviewer BfMQ22’s comment that the experimental results on NLP tasks would further strengthen the significance of our work, we conducted additional experiments involving text classification. Details can be found in our response to reviewer BfMQ22.

---

> ### Author Response · Authors · 2023-11-20
> **Kindly Reminder**
>
> As the rebuttal period draws to a close, we would really appreciate receiving your valuable input.
> We are encouraged by the positive and constructive dialogue with reviewer BfMQ (please see our
> posted exchange). Any suggestions that you may have would greatly contribute to further enhancing
> our work!

---

> > ### Author Response · Authors · 2023-11-22
> > **Kindly Reminder**
> >
> > As the rebuttal window is coming to a close, a kind reminder to please let us know if there is any other clarification that we could provide… thanks!

---

### Author Response · Authors · 2023-11-15
**General Response to all readers**

General response to all reviewers:

We thank the reviewers for their efforts and for the feedback they provided.  For convenience, we first summarize and briefly address the main concerns of each reviewer, and then proceed with point-by-point responses to all the comments.

(1) Reviewer yWNZ02 expressed concerns about the performance of our method when deployed with more complicated networks, and suggested adding baselines to the experiments. In response, we conducted supplementary experiments which demonstrate that the proposed method outperforms competing techniques in all the considered settings.

(2) Reviewer V18X30 questions the contributions of the manuscript. In response, we summarize the novel contributions and compare/contrast them to the related work.

(3) Reviewer BfMQ22 primarily expressed concerns regarding applicability of our method to settings beyond those considered in the paper. While the presented experimental results are focused on image classification tasks, our method is model-agnostic and can be used in any classification tasks (not limited to images) with any model architecture.

We have attempted to respond to reviewers’ questions as soon as possible, making better use of the limited rebuttal period. Thank you!

Respectfully,

Authors

---

### Meta-Review · Area_Chair_hu2M · 2023-12-11

**Metareview:**

This paper develops a client sampling scheme for federated learning based on hierarchical clustering. The approach aims to address the data heterogeneity challenge in FL by sampling clients in a way to "debias" the results. The approach is novel, some theoretical convergence guarantees are provided, and the reported experimental results on small datasets are promising. I want to acknowledge the concerns communicated directly to the AC by the authors, and in particular acknowledge agreement that Appendix A.9 does indeed discuss extensions to algorithms besides SGD.

The paper could be strengthened in a few ways. First, multiple reviewers noted that the experimental results, while promising, are limited in terms of size and scope. Reviewer yWNZ pointed out the FedCor baseline, and Reviewer BfMQ requested experiments on additional modalities, leading to the addition of a comparison with FedCor and additional experiments on a small text classification workload. The inclusion of these experiments is acknowledged and appreciated. Still, client sampling is most commonly motivated by settings with a very large number of clients (many thousands or more), while the experiments in this study focus on a few 10's of clients, leading to questions about scalability and effectiveness at scale.

Next, Reviewer V18X pointed out that the approach does not directly address privacy concerns. The response from the authors was that clients send the same information as they would normally, namely updated model parameters / gradients. However, this overlooks one important aspect of practical FL deployments. It has been previously shown that gradient inversion attacks can be applied to model updates from individual clients to recover the client's training samples, thereby violating their privacy. To avoid this, it is usually recommended to use a secure aggregation method to implement the averaging (e.g., line 17 in Algorithm 1 from the paper); see *Towards Federated Learning at Scale" by Bonawitz et al., 2019. In this way, the server only has access to information aggregated over multiple clients, and no party has visibility into the updates from any individual client. However, to implement HiCS-FL the server needs access to information from individual clients.

A more detailed discussion of these issues and tradeoffs involved would strengthen the paper.

**Justification For Why Not Higher Score:**

Current experiments focus on small-scale problems while the motivation for the approach is larger settings. And it isn't clear that the proposed approach is compatible with privacy requirements, which are an essential tenet of federated learning.

**Justification For Why Not Lower Score:**

N/A

---

### Decision · Program_Chairs · 2024-01-16

Reject